# *Syn*DLP is a dynamin-like protein of *Synechocystis* sp. PCC 6803 with eukaryotic features

Lucas Gewehr[1,8], Benedikt Junglas [2,3,8], Ruven Jilly[1], Johannes Franz[4], Wenyu Eva Zhu[1], Tobias Weidner [5], Mischa Bonn [4], Carsten Sachse [2,3,6] ✉ & Dirk Schneider [1,7] ✉

Dynamin-like proteins are membrane remodeling GTPases with well-understood functions in eukaryotic cells. However, bacterial dynamin-like proteins are still poorly investigated. *Syn*DLP, the dynamin-like protein of the cyanobacterium *Synechocystis* sp. PCC 6803, forms ordered oligomers in solution. The 3.7 Å resolution cryo-EM structure of *Syn*DLP oligomers reveals the presence of oligomeric stalk interfaces typical for eukaryotic dynamin-like proteins. The bundle signaling element domain shows distinct features, such as an intramolecular disulfide bridge that affects the GTPase activity, or an expanded intermolecular interface with the GTPase domain. In addition to typical GD-GD contacts, such atypical GTPase domain interfaces might be a GTPase activity regulating tool in oligomerized *Syn*DLP. Furthermore, we show that *Syn*DLP interacts with and intercalates into membranes containing negatively charged thylakoid membrane lipids independent of nucleotides. The structural characteristics of *Syn*DLP oligomers suggest it to be the closest known bacterial ancestor of eukaryotic dynamin.

Cells employ membrane remodeling proteins for diverse physiological processes, including endocytosis, exocytosis, membrane fusion and fission, and membrane repair[1–3]. Efficient membrane repair mechanisms are also indispensable for cells to cope with membrane damage, finally ensuring cell survival. Membrane remodeling is vital to maintain cellular compartmentalization by membrane-enclosed organelles in eukaryotes, as well as the maintenance of prokaryotic membrane systems. In fact, within the last decade, proteins have been identified in prokaryotes, which are involved in membrane repair and/or dynamics. Many of these proteins are homologs of proteins previously assumed to be eukaryotic inventions. Examples are the proteins FtsA, FtsZ and ZipA, which mediate membrane constriction during bacterial cytokinesis, and are homologous to the eukaryotic proteins tubulin, actin, and MAP-Tau[4–8].

Similarly, dynamins and dynamin-like proteins (DLPs) were originally assumed to be eukaryotic inventions until, in 1999, a bioinformatic study predicted the existence of bacterial DLPs (BDLPs)[9]. In eukaryotes, DLPs are involved in various membrane remodeling processes, such as endocytosis or fission and fusion of organelle membranes[10–17]. Unlike small (Ras-like) GTPases, DLPs are large mechanochemical GTPases with a molecular mass >60 kDa that use the energy gained via GTP hydrolysis for membrane binding and/or remodeling[18]. When bound, GTP is hydrolyzed, and cleavage of the γ-phosphate induces conformational changes leading to movements

[1]Department of Chemistry, Biochemistry, Johannes Gutenberg University Mainz, Mainz, Germany. [2]Ernst Ruska-Centre for Microscopy and Spectroscopy with Electrons (ER-C-3): Structural Biology, Jülich, Germany. [3]Institute for Biological Information Processing (IBI-6): Cellular Structural Biology, Jülich, Germany. [4]Max Planck Institute for Polymer Research, Ackermannweg 10, 55128 Mainz, Germany. [5]Department of Chemistry, Aarhus University, Langelandsgade 140, 8000 Aarhus C, Denmark. [6]Department of Biology, Heinrich Heine University, Universitätsstr. 1, Düsseldorf, Germany. [7]Institute of Molecular Physiology, Johannes Gutenberg University Mainz, Mainz, Germany. [8]These authors contributed equally: Lucas Gewehr, Benedikt Junglas. ✉e-mail: c.sachse@fz-juelich.de; Dirk.Schneider@uni-mainz.de

of conserved DLP domains relative to each other. The inorganic phosphate is then released, resulting in a GDP-bound DLP. Upon GDP dissociation, a new GTP can bind to the nucleotide-free enzyme to start a new cycle[19].

While DLP family members are typically not highly conserved on the sequence level, with the exception of the GTPase domain (GD), the resolved structures reveal a conserved modular arrangement of all DLPs: The globular GD at the protein's N-terminus is typically followed by an α-helical bundle signaling element (BSE) or neck domain that connects the GD to an α-helical stalk or trunk domain. Most DLPs additionally have membrane interaction domains (MIDs) of varying designs[20,21].

The GD, the only structural element of DLPs that is conserved at the sequence level, is characterized by a low μM nucleotide binding-affinity plus a relatively high basal GTPase activity, at least when compared to other GTPases. E.g., Ras-like GTP-binding proteins require GTPase-activating proteins (GAPs) for nucleotide hydrolysis, whereas DLPs operate independently of GAPs[22–26]. Yet, the GTPase activity of DLPs typically increases when DLP monomers oligomerize, as the GTPase activity is regulated by intermolecular GD contacts leading to a head-to-head dimerization of adjacent GDs. E.g., dynamin dimers/tetramers show a basal GTPase activity of ~1 min$^{-1}$ when free in solution. Yet, upon binding to membranes, dynamins oligomerize on the membrane surface, resulting in GTPase-activating GD contacts and release of auto-inhibitory GD-MID contacts[22,23,27]. This can finally increase the GTPase activity >100-fold[28–30].

Functionally, the majority of eukaryotic DLPs can be subdivided into either (i) membrane fission or (ii) membrane fusion DLPs[31]. Dynamin, the founder and namesake of the dynamin superfamily, is a fission DLP, involved in clathrin-mediated endocytosis[10,11,32]. Other fission DLPs, such as Drp1 and DRP3A, act on cell organelles, involving mitochondria and peroxisomes[12,13]. Conversely, fusion DLPs, such as mitofusin, OPA1 or atlastin, fuse the mitochondrial outer membrane, the mitochondrial inner membrane or the endoplasmic reticulum membrane, respectively[14,15,33]. The fusogenic DLP Fzl is involved in the remodeling of thylakoid membranes (TMs) in chloroplasts[16,17,34].

About two decades ago, a BDLP was characterized for the first time in the cyanobacterium *Nostoc punctiforme* (*Np*BDLP)[35]. While the physiological functions of BDLPs are mostly enigmatic, a BDLP of *Bacillus subtilis* (*Bs*DynA) has recently been shown to be involved in membrane stabilization and defense against phage infection[36]. For other BDLPs, several physiological functions were proposed, such as vesicle release or biogenesis (*Ec*LeoA, *Ms*IniA) and crucial involvement in cytokinesis (*Sv*DynA/B)[37–40]. As their eukaryotic counterparts, also BDLPs appear to be either involved in membrane fission or fusion, and, e.g., MsIniA has a membrane fission activity whereas *Bs*DynA and *Cj*-DLP1/2 are able to fuse membranes, at least in vitro[41–43].

Recently, a BDLP has also been identified in the genome of the cyanobacterium *Synechocystis* sp. PCC 6803 (*Syn*DLP)[44]. In contrast to other prokaryotes, cyanobacteria typically contain an uncommon second, completely separated internal membrane system besides the cytoplasmic membrane, the TMs where the photosynthetic light reaction takes place. TMs have a rather unique lipid composition, with the two neutral galactolipids monogalactosyldiacylglycerol (MGDG) and digalactosyldiacylglycerol (DGDG) being the major membrane lipids, plus the negatively charged lipids sulfoquinovosyldiacylglycerol (SQDG) and the only phospholipid phosphatidylglycerol (PG)[45,46]. There are still many open questions concerning the biosynthesis of TMs, e.g., whether the TMs are completely assembled de novo or not[47,48]. Due to the photosynthetic light reaction, TMs are highly vulnerable to light stress and are continuously remodeled[49], and thus, proteins mediating membrane remodeling and/or repair via membrane fusion and fission are required. As the involvement of DLPs in membrane dynamics and/or repair is well-established in eukaryotes, it is feasible to also assume involvement of *Syn*DLP in similar processes in *Synechocystis* sp. PCC 6803. Yet, currently *Syn*DLP solely is a predicted DLP and no information as to its structure and function are available.

Here, we show that purified *Syn*DLP is a bona fide DLP that specifically interacts with negatively charged TM lipids. Furthermore, *Syn*DLP assembles into ordered high molecular mass oligomers. The structure of *Syn*DLP oligomers reveals oligomeric interfaces in the stalk domain typical for eukaryotic fission DLPs. Based on an analysis of an intramolecular disulfide bridge stabilizing the BSE and an assembly-impaired *Syn*DLP variant, we propose a GTPase activity-regulating function for the BSE domain. The interaction of *Syn*DLP with negatively charged lipid headgroups is nucleotide-independent.

## Results

### Cryo-EM structure of *Syn*DLP oligomers

Recently, in a bioinformatic analysis, the *orf slr0869* of the *Synechocystis* sp. PCC 6803 genome has been identified to encode a DLP (*Syn*DLP)[44]. A sequence alignment and homology search revealed the presence of a GD with the typical GTP-binding motifs, including the G1-motif/P-loop, the G2-motif/switch I, the G3-motif/switch II and the G4-motif, all hallmarks of dynamin-like GTPases[18,20,21]. *Syn*DLP appears not to be essential in the cyanobacterium under standard growth conditions (Supplementary Fig. 1a) albeit it is expressed in vivo (Supplementary Fig. 1b). A typical sodium dodecyl sulfate polyacrylamide gel electrophoresis (SDS-PAGE) analysis of isolated, heterologously produced *Syn*DLP is shown in Supplementary Fig. 2. Analytical size exclusion chromatography revealed that *Syn*DLP forms oligomers in solution, a feature typically observed when DLPs bind nucleotides or membranes (Fig. 1a). Yet, in contrast to other BDLPs, *Syn*DLP forms high-molecular mass oligomers already in complete absence of an externally added membrane template and/or nucleotides. In a sedimentation assay, the addition of GTP or GDP led to only marginal changes in the sedimentation behavior of *Syn*DLP, indicating only a minor shift to larger structures in the presence of GTP and to smaller structures after GDP addition (Supplementary Fig. 3). When we visualized *Syn*DLP by cryo-EM using prepared plunge-frozen vitrified specimen, we observed short oligomeric filaments of bent half-moon shape (Fig. 1b) with typical lengths of about 100 nm and a curvature radius of ~50 nm. Noteworthy, in contrast to biochemical assays, EM micrographs were acquired in the absence of NaCl, as the oligomeric filaments appeared longer and more defined under these conditions (Supplementary Fig. 4) and therefore more suitable for structural analysis. Class averages of these elongated oligomers revealed a width of 150 Å and a repeating unit every 60 Å along the oligomer (Fig. 1c). Using a segmented single-particle analysis workflow (for details see Supplementary Fig. 5), we determined the structure of *Syn*DLP oligomers at an overall resolution of 3.7 Å (Fig. 1d, Table 1). The local resolution varied from 3.0 to 3.5 Å at the stalk domain to 5.0 to 7.0 Å at the GD, presumably due to tighter contacts between the well-packed stalk domains and looser contacts between adjacent GDs. The last 19 residues at the C-terminus could not be resolved in the cryo-EM density, and thus, the final refined atomic model includes amino acids (aa) 1-793 of *Syn*DLP. Based on the determined cryo-EM map, we built a model of 8 *Syn*DLP monomers, which represents a smaller part of the segmented *Syn*DLP oligomer structure. When the determined octamer is extended, matching the observed dimensions of the 2D class averages, it will consist of ~40–50 monomers. The curvature radius of 50 nm of the *Syn*DLP oligomers is rather high compared to similar assemblies of other DLPs that typically assemble in the presence of membranes and/or nucleotides. Here, the curvature radius is usually in the range of 13 to 26 nm (reviewed here[20]). However, other DLP assemblies also show a low curvature comparable to *Syn*DLP, e.g., Drp1[50].

The *Syn*DLP structural model revealed a domain architecture typical for eukaryotic fission DLPs[27,50] (Fig. 1e–g). The GD is flanked by parts of the BSE domain (N-terminally by BSE1, C-terminally by BSE2)

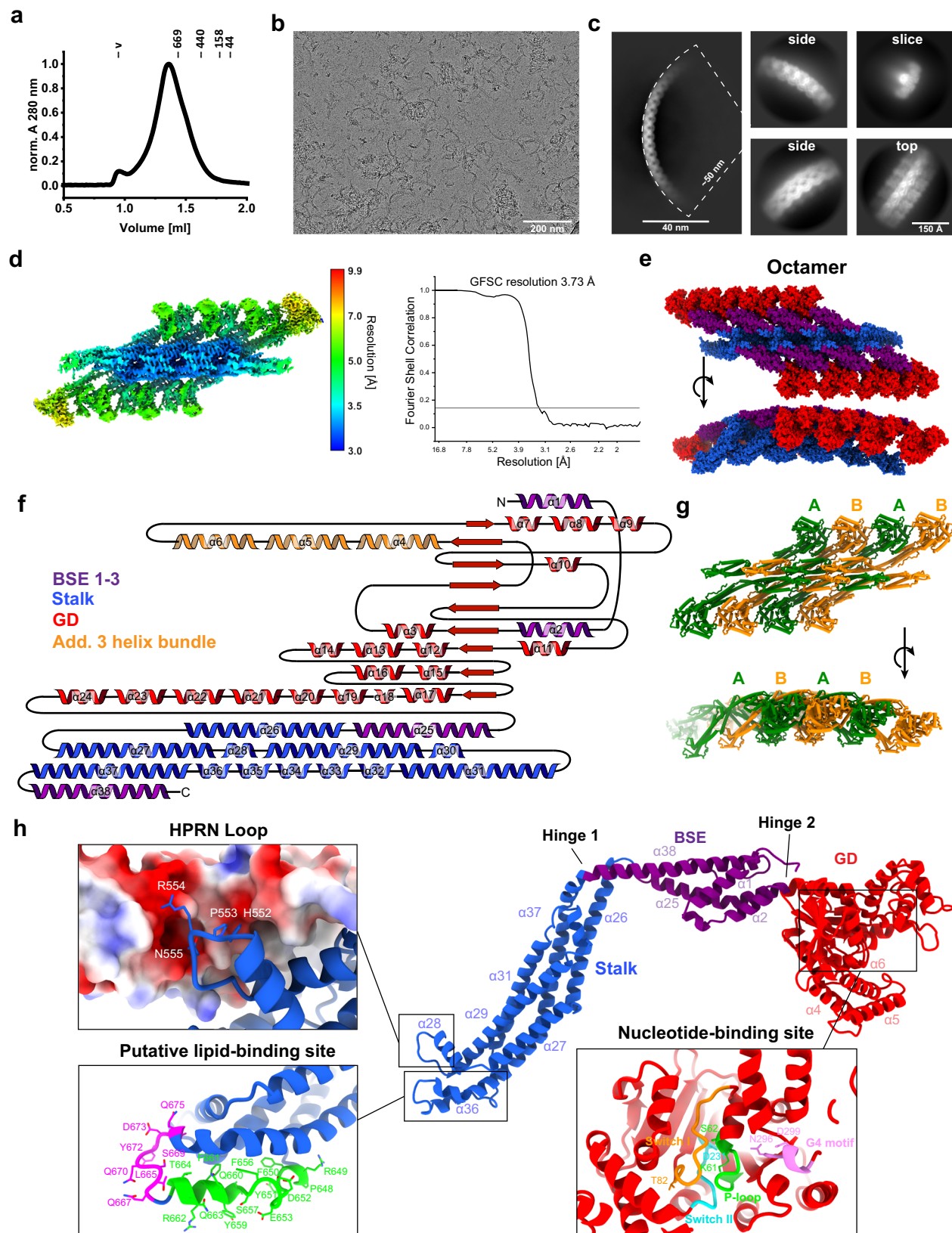

followed by a stalk domain. The BSE3 domain part is located C-terminally to the stalk (Fig. 1f, Supplementary Fig. 6). The monomer model revealed a globular GD, linked via a flexible hinge region (hinge 2) to the mainly α-helical BSE domain, which is connected to an α-helical stalk via hinge 1 (Fig. 1h). The oligomer structure showed a stalk backbone connected via a complex interaction network

(Supplementary Fig. 7) from which the BSE domains and the GDs protrude laterally outwards (Fig. 1e). The projected center of the stalk domains lies on a curve conferring the curvature of the *Syn*DLP oligomers. A loop consisting of an HPRN motif, which mediates critical contacts to neighboring monomers via electrostatic interactions (see below), as well as putative lipid-binding sites are located at the tip of

**Fig. 1 | Cryo-EM structure of oligomeric *Syn*DLP. a** Analytical size exclusion chromatography of *Syn*DLP (black) revealed the formation of oligomeric structures that are larger than common standard proteins (standard proteins' peak positions and molecular masses in kDa are indicated, v = void volume). Absorption values at 280 nm were normalized (0–1). **b** Cryo-EM micrograph of *Syn*DLP oligomers. The data set of 8322 micrographs was measured one time (no independent replicates). **c** Class averages of *Syn*DLP oligomers with an enlarged oligomer side view including the curvature radius (left) and focused views of side, top, and slice view. **d** Local resolution map and FSC curve (with auto-masking) of the *Syn*DLP reconstruction. **e** Model of a *Syn*DLP octamer with GD (red), BSE1-3 (purple) and stalk

(blue). **f** Secondary structure topology plot of *Syn*DLP with additional α-helices of the enlarged *Syn*DLP GD colored in orange. **g** Model of a *Syn*DLP octamer highlighting the sequential arrangement of monomers within the oligomer. Alternating monomers are colored in green and orange, respectively. **h** Model of the *Syn*DLP monomer, including structural features. GD, BSE and stalk are colored as in (**e**). Zoomed insets show the HPRN loop with the electrostatic surface of neighboring monomers, two putative lipid-binding sites at the tip of the stalk colored in magenta and green, respectively, and conserved motifs in the nucleotide-binding site of the GD: P-loop, switch I, switch II and G4 motif colored in green, orange, cyan and pink, respectively.

the stalk (Fig. 1h). Note that molecular details of the GD are more difficult to annotate due to the intermediate resolution of this part.

### *Syn*DLP is a potential bacterial ancestor of eukaryotic DLPs

The *Syn*DLP monomer structure revealed typical DLP features (Fig. 1h). The structure of *Syn*DLP oligomers resembles the assembly of eukaryotic fission DLPs via a complex network of interactions in the stalk domain, mainly mediated by three distinct interfaces (Supplementary Fig. 7b–d). Interestingly, these interfaces are conserved in some

### Table 1 | Cryo-EM data collection and refinement statistics

| Data collection and processing | *Syn*DLP |
|---|---|
| Number of movies collected | 8322 |
| Magnification | ×49,000 |
| Voltage [kV] | 200 |
| Electron exposure [e⁻/Å²] | 26.5 |
| Underfocus range [μm] | 2.0–4.0 |
| Physical pixel size [Å] | 1.737 |
| Detector | Gatan K3 |
| Symmetry imposed | C2 |
| Final no. of particles | 977,199 |
| Global map resolution [Å, FSC = 0.143] | 3.7 |
| Local map resolution range [Å] | 3.0–9.9 |
| Initial model used (PDB code) | – |
| Model refinement | *Syn*DLP |
| Model resolution | 3.7 |
| CC mask | 0.60 |
| CC box | 0.79 |
| CC peaks | 0.53 |
| CC volume | 0.5.9 |
| Map sharpening B-factor [Å²] | –184 |
| Model composition | |
| Nonhydrogen atoms | 101,416 |
| Protein residues | 6344 |
| RMSDs | |
| Bond lengths [Å] | 0.004 |
| Bond angles [°] | 0.926 |
| Validation | |
| MolProbity score | 1.59 |
| Clashscore | 8.56 |
| Rotamer outliers [%] | 0.31 |
| Ramachandran plot | |
| Favored [%] | 97.35 |
| Allowed [%] | 2.65 |
| Disallowed [%] | 0.00 |
| Deposition IDs | |
| EMDB | 14993 |
| PDB | 7ZW6 |

eukaryotic DLPs[27,50,51], yet, have not been reported in any prokaryotic DLP structure thus far (Supplementary Fig. 8a). A surface conservation plot of *Syn*DLP revealed highly conserved residues in the nucleotide-binding pocket of the GD and laterally at the stalk (Fig. 2a, red-violet, Supplementary Fig. 9), which correspond to the conserved GD motifs and the oligomerization interfaces 2 and 3 (Fig. 1h, Supplementary Fig. 7c, d). Structural analysis of isolated *Syn*DLP domains showed an increased size of the GD compared to other DLP GDs (Fig. 2b). In fact, the *Syn*DLP GD is >100 aa larger than typical dynamin-like GDs as it contains additional loops and α-helices (Supplementary Fig. 10a). In a structural alignment with eukaryotic and bacterial DLP representatives, the estimated root-mean-square deviation (RMSD) of Cα positions at 8.5–10.2 Å indicate a closer relation of *Syn*DLPs α-helices in the stalk domain to the eukaryotic DLP members Dyn3 and MxA as opposed to other bacterial members at >14 Å (Fig. 2c), which is in line with the stalk-mediated assembly thus far observed solely in eukaryotic representatives (Supplementary Fig. 8a). The close relation of *Syn*DLP to eukaryotic DLPs is further supported by structural alignments of isolated BSE domains (Supplementary Fig. 10b). Taken together, *Syn*DLP is a prokaryotic DLP with an enlarged GD and shows structural features not observed in BDLPs before. Thus, *Syn*DLP is the closest known bacterial ancestor of a class of eukaryotic DLPs, such as dynamin or MxA.

### An intramolecular disulfide bridge in the BSE domain influences the *Syn*DLP GTPase activity

The *Syn*DLP structure indicated the formation of an intramolecular disulfide bridge between C8 and C777 that covalently connects the BSE1 and BSE3 domains (Fig. 3a). In fact, the migration behavior of *Syn*DLP in SDS-PAGE analysis changed depending on the DTT concentration (Fig. 3b): at low DTT concentrations, i.e., when the disulfide bridge is established, *Syn*DLP ran at an increased apparent molecular mass whereas it ran at the predicted molecular mass in the presence of DTT. To study the structural impact of the intramolecular disulfide bridge, a *Syn*DLP variant was generated by mutating one of the involved cysteines (C777) to alanine. In the resulting protein (*Syn*DLP_{C777A}) the formation of the disulfide bridge was prevented (Supplementary Fig. 2). When analyzed via size exclusion chromatography, *Syn*DLP_{C777A} behaved like *Syn*DLP wt (Fig. 3c), and thus, the C777A mutation apparently does not affect *Syn*DLP oligomer formation. In addition, the mutant protein seemed to be correctly folded at the secondary structure level as indicated by circular dichroism (CD) spectra of *Syn*DLP wt and *Syn*DLP_{C777A}, which were identical at 20 °C (Fig. 3d). Yet, the mutation affected the thermodynamic stability of *Syn*DLP. The thermal stability of *Syn*DLP wt and *Syn*DLP_{C777A} was investigated by thermal denaturation of the proteins. Unfolding of the secondary structure was monitored via recording CD spectra in the far UV at increasing temperatures (Fig. 3e). The spectra were dominated by minima at 208 and 222 nm, due to the high content of α-helices. While from the melting curve of *Syn*DLP wt (Fig. 3e, black) a T_{m, CD} of 49.5 ± 0.3 °C was calculated, the T_{m, CD} of the *Syn*DLP_{C777A} variant (Fig. 3e, red) was determined to be 43.7 ± 0.3 °C (Table 2), showing that *Syn*DLP_{C777A} is less stable than the wt. This observation was further elucidated via an ANS fluorescence thermal shift assay (FTSA), which

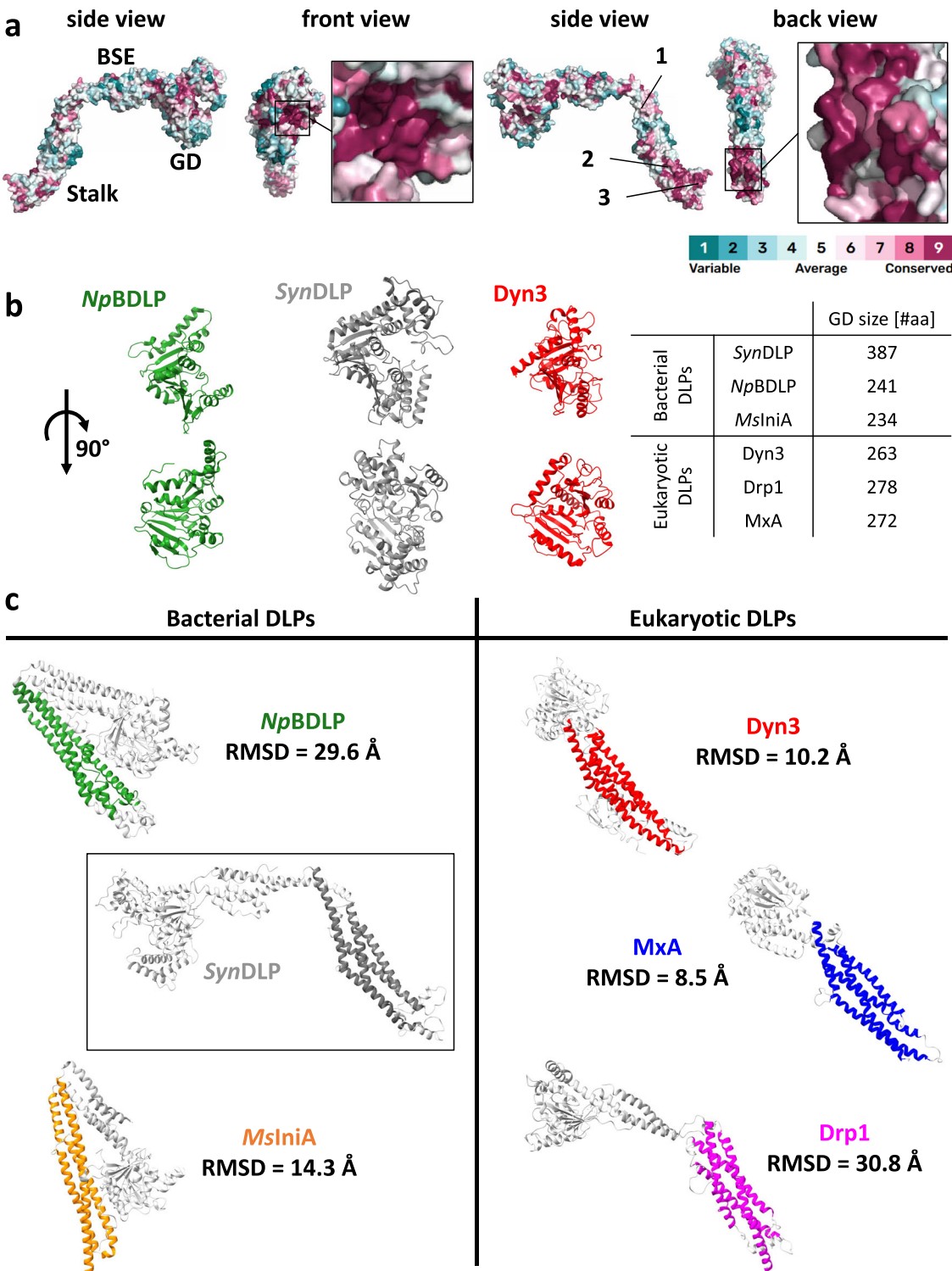

**Fig. 2 | Structural comparison of *Syn*DLP with other bacterial and eukaryotic DLP structures. a** A surface conservation plot showed amino acids in the *Syn*DLP structure that are conserved in other DLP sequences. The surface conservation plot was produced using ConSurf[101–105]. The color key ranges from 1 (cyan, variable regions) to 9 (red-violet, conserved regions). The evaluation of single residues conservation in the primary sequence is shown in Supplementary Fig. 9. The position of GD, BSE, and stalk are highlighted in the first side view. The oligomerization interfaces 1–3 are labeled in the second side view. **b** Structural side-by-side comparison of dynamin-like GDs from *Syn*DLP (gray) with a bacterial (*Np*BDLP,

PDB: 2J69, green) and a eukaryotic representative (Dyn3, PDB: 5A3F, red). The table summarizes the size of different GDs (*Syn*DLP, *Np*BDLP (PDB: 2J69), *Ms*IniA (PDB: 6J73), Dyn3 (PDB: 5A3F), Drp1 (PDB: 5WP9) and MxA (PDB: 3SZR)). **c** Structural arrangement of α-helices in the stalk domains of *Syn*DLP (gray) and four other DLPs was compared by structural alignment of the domains leading to Cα-RMSD values larger than 10. The areas used for the structural alignments are highlighted in green (*Np*BDLP), orange (*Ms*IniA), red (Dyn3), blue (MxA) or magenta (Drp1) and the remaining structural elements not used for the alignment are colored in light gray. For details about the number of aligned residues, see Supplementary Fig. 10c.

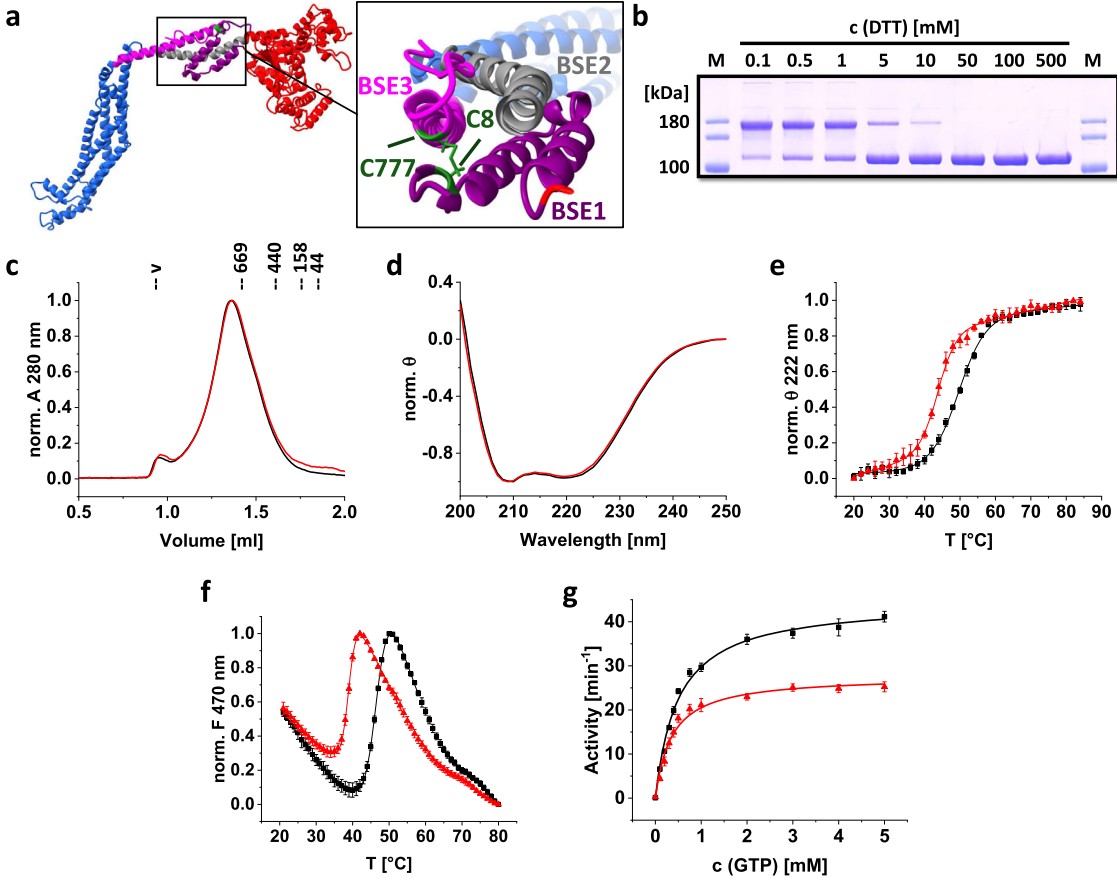

**Fig. 3 | Structural stabilization and enhanced GTPase activity by an intramolecular disulfide bridge in the BSE domain. a** Ribbon representation of a *Syn*DLP monomer. C8 and C777, forming an intramolecular disulfide bridge that connects BSE1 and BSE3, are highlighted in green. BSE1, which consists of two α-helices, BSE2 and BSE3 are colored purple, gray and magenta, respectively. GD colored in red, stalk in blue. **b** *Syn*DLP was incubated with different DTT concentrations (the final concentrations after adding SDS sample buffer are given). The amount of *Syn*DLP with an intact disulfide bridge depended on the DTT concentration (band with intact bridge at ~180 kDa, band with reduced cysteines at ~100 kDa). M = marker. Representative gel from two independent experiments with the same result. **c** Analytical size exclusion chromatography of *Syn*DLP$_{C777A}$ (red) compared to *Syn*DLP wt (black) revealed the formation of oligomeric structures for both proteins. A$_{280}$ values were normalized (0–1) for better comparison. **d** CD spectra of 1 μM *Syn*DLP wt (black) and *Syn*DLP$_{C777A}$ (red). Mean of three measurements is

shown. For comparison, the spectra were normalized (θ value at 250 nm set to 0, minimum θ set to −1). **e** The thermal stability of *Syn*DLP (black) and *Syn*DLP$_{C777A}$ (red) was monitored using CD spectroscopy. The ellipticities at 222 nm were plotted against the temperature, normalized (0–1) and fitted with an adapted Boltzmann fit (Eq. (3)). The fit curves are shown as lines. Mean of independent measurements (*n* = 3) and error bars (S.D.) are shown. **f** Measured ANS-FTSA of *Syn*DLP wt (black) and *Syn*DLP$_{C777A}$ (red). ANS fluorescence intensities at 470 nm were plotted against the temperature and normalized (0–1) for better comparison. A temperature range capturing the transition phase was fitted with an adapted Boltzmann fit (Eq. (4)). Fit curves are shown as lines. The mean of three independent experiments and error bars (S.D.) are displayed. **g** The GTPase activity of *Syn*DLP wt (black) and *Syn*DLP$_{C777A}$ (red) were measured in a continuous, regenerative, coupled assay. The mean and error bars (S.D.) of three independent experiments are shown. The data points were fitted using Eq. (2).

monitors changes in the tertiary and quaternary structure of proteins and thus complements the CD measurements. The T$_{m, ANS-FTSA}$ was shifted by ~8 °C from 46.7 ± 0.1 °C to 39.0 ± 0.1 °C for the mutant (Fig. 3f, Table 2), which further confirmed the lowered thermodynamic stability of *Syn*DLP$_{C777A}$ compared to *Syn*DLP wt.

As DLPs catalyze the hydrolysis of GTP to GDP and P$_i$, the GTPase activity of *Syn*DLP and *Syn*DLP$_{C777A}$ was determined next via a continuous, regenerative, coupled GTPase assay. Typically, the GTPase assay was performed with 0.5 μM protein, as *Syn*DLP showed a concentration-dependent GTPase activity at low protein concentra-

tions yet reaching a plateau at protein concentrations >0.3 μM (Supplementary Fig. 11a), and in the presence of the monovalent cations Na$^+$ and K$^+$, which were shown to activate the GTP hydrolysis catalyzed by DLPs[22]. E.g., the omission of NaCl led to a clear decrease of the *Syn*DLPs GTPase activity (Supplementary Fig. 11b). A mutation of a conserved residue in the P-loop (K61A) considerably reduced the GTPase activity, as previously observed for dynamin and other DLPs, which demonstrated a related GTPase mechanism (Supplementary Fig. 11c). The GTPase activities of *Syn*DLP wt and C777A followed a typical Michaelis-Menten kinetic (Fig. 3g), and consequently the data were fitted with a hyperbolic curve (Eq. (2)). The turnover rate (k$_{cat}$) and the Michaelis-Menten constant (K$_m$) were determined, as summarized in Table 3. Apparently, with a k$_{cat}$ of 44.6 ± 1.0 min$^{-1}$ *Syn*DLP had a relatively high basal GTPase activity (Fig. 3g, black). Yet, the turnover rate of the *Syn*DLP$_{C777A}$ mutant was significantly reduced by almost 40% compared to the wt (Fig. 3g, red, Table 3). Next, GTP binding affinities of *Syn*DLP wt and C777A were determined via fluorescence anisotropy measurements using the GTP analog Mant-GTP (Supplementary Fig. 12). Based on this assay, both proteins have

**Table 2 | Transition temperatures of *Syn*DLP variants determined via CD spectroscopy and ANS-FTSA**

| | T$_{m, CD}$ [°C] | T$_{m, ANS-FTSA}$ [°C] |
|---|---|---|
| *Syn*DLP | 49.5 ± 0.3 | 46.7 ± 0.1 |
| *Syn*DLP$_{C777A}$ | 43.7 ± 0.3 | 39.0 ± 0.1 |
| *Syn*DLP$_{HPRN-AAAA}$ | 46.4 ± 0.1 | 44.9 ± 0.1 |

similar nucleotide-binding affinities in the three-digit nanomolar range. As both proteins showed no residual (measurable) GTPase activity at 0 mM GTP (negative control without nucleotide; first data point in each curve) although a GTP regenerating system was added, we conclude that no significant amount of GTP/GDP has been co-purified with the proteins. This is supported by the low $A_{260}/A_{280}$ ratio of 0.7 determined via absorption spectroscopy using purified *Syn*DLP, a value indicating that no nucleotides were bound.

### Uncommon GD contacts enhance the *Syn*DLP GTPase activity

In the *Syn*DLP oligomer structure we observed an expanded inter-molecular GD-BSE1 interface (Fig. 1e, Supplementary Fig. 8b). The role of the intermolecular GD-BSE1 interface in *Syn*DLP oligomers was addressed via analysis of an assembly-defective *Syn*DLP variant. *Syn*DLP oligomerizes via an intricate interaction network in the stalk

domain by contacts of defined interfaces as already mentioned (Supplementary Fig. 7). As a general feature, structurally related DLPs form a stable dimer via interface 2. The basic dimeric unit can tetramerize or further oligomerize via interfaces 1 and 3. The latter is non-symmetric and mediates lateral contacts between parallely oriented stalks. Following previous studies on diverse DLPs where oligomerization was disturbed by mutations, we replaced the residues [552]HPRN[555] in a highly conserved loop at the tip of the stalk domain that is part of oligo-merization interface 3 (Fig. 1h, Supplementary Fig. 13a)[52–56] by four alanines.

The migration behavior of *Syn*DLP_{HPRN-AAAA} on an SDS PAGE gel (Fig. 4a) and the wt-like shape of the CD spectrum (Fig. 4b) indicated proper secondary structure formation as well as the formation of the stabilizing disulfide bridge within the BSE of the mutated protein. Yet, analytical size exclusion chromatography of *Syn*DLP_{HPRN-AAAA} revealed an apparent MW of ~174 kDa, indicating formation of dimers (Fig. 4c, red), which was previously also observed for other DLPs carrying the equivalent mutations[52,53,56]. The hampered oligomerization was also confirmed by cryo-EM micrographs (Supplementary Fig. 13b+c). Thermal denaturation of *Syn*DLP and *Syn*DLP_{HPRN-AAAA} was monitored via CD spectroscopy and ANS-FTSA to investigate the impact of oli-gomerization on the thermodynamic stability. With both methods, a slight decrease of the T_m of *Syn*DLP_{HPRN-AAAA} was observed, which indicated that oligomerization promotes the thermodynamic stability (Fig. 4d, e, Table 2). Compared to *Syn*DLP wt (Fig. 4e, black), the ANS-

**Table 3 | Kinetic parameters describing the GTPase activity of *Syn*DLP variants**

|  | $k_{cat}$ [min$^{-1}$] | $K_m$ [mM] |
|---|---|---|
| *Syn*DLP | 44.6 ± 1.0 | 0.50 ± 0.04 |
| *Syn*DLP_{C777A} | 27.7 ± 0.8 | 0.35 ± 0.04 |
| *Syn*DLP_{HPRN-AAAA} | 17.0 ± 0.2 | 0.33 ± 0.02 |

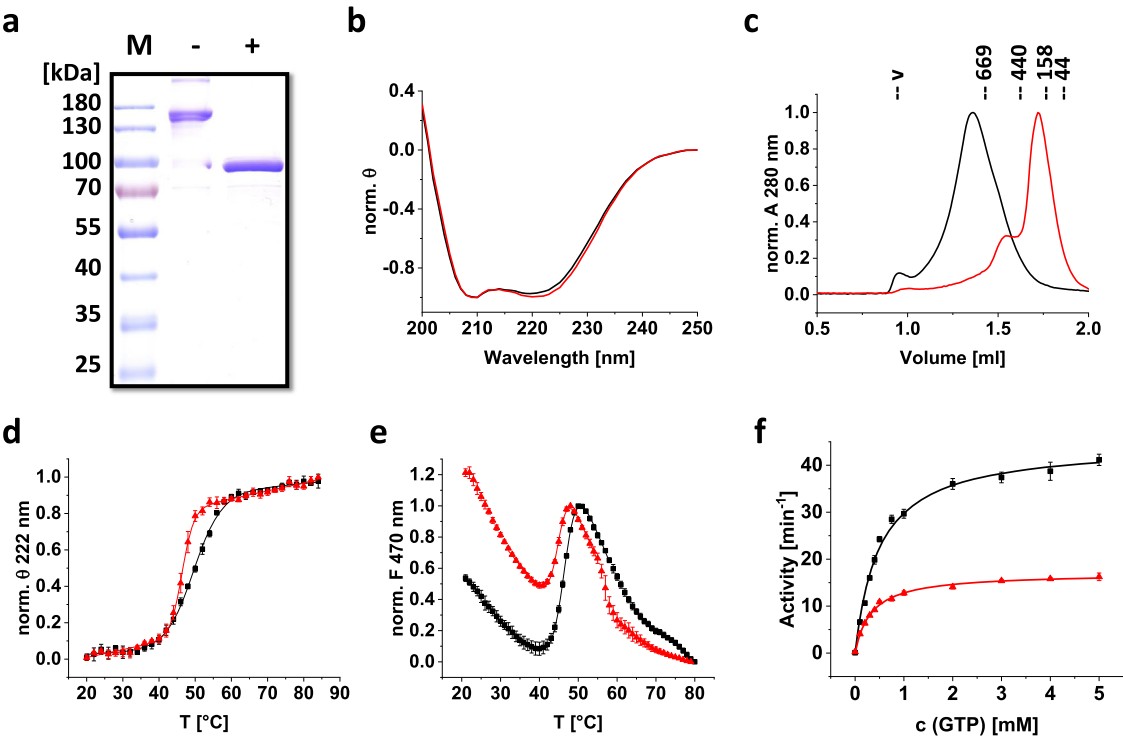

**Fig. 4 | Biochemical analysis of *Syn*DLP_{HPRN-AAAA}. a** Purified *Syn*DLP_{HPRN-AAAA} was analyzed via SDS-PAGE in the presence of 0.1 mM DTT (lane −) or 100 mM DTT (lane +). The calculated molecular mass of the protein is 93 kDa. The SDS-PAGE analysis revealed a single band at ~100 kDa (lane +) without showing further protein bands, thus the protein was ≥95% pure. M = marker. Representative gel of two independent experiments showing the same results. **b** CD spectra of *Syn*DLP wt (black) and *Syn*DLP_{HPRN-AAAA} (red). The mean of three measurements is shown. The spectra were normalized (θ value at 250 nm set to 0, minimum θ set to −1) for better comparison. **c** Analytical size exclusion chromatography of *Syn*DLP wt (black) and *Syn*DLP_{HPRN-AAAA} (red). The elution peak positions of the standard proteins and the corresponding molecular masses are indicated (v = void volume). A_{280} values were normalized (0–1) to better compare the chromato-grams. **d** CD spectra of *Syn*DLP wt (black) and *Syn*DLP_{HPRN-AAAA} (red) were

recorded at increasing temperatures. The ellipticities at 222 nm were plotted against the temperature, normalized (0–1) and fitted with an adapted Boltzmann fit (Eq. (3)). The fit curves are displayed as lines. Mean of three independent experiments and error bars (S.D.) are shown. **e** ANS-FTSA measurements of *Syn*DLP wt (black) and *Syn*DLP_{HPRN-AAAA} (red). ANS fluorescence intensities at 470 nm were plotted against the temperature and normalized (minimum set to 0, main peak set to 1). The temperature range that captured the transition phase was fitted with an adapted Boltzmann fit (Eq. (4)). Fit curves are shown as lines. Error bars represent S.D., *n* = 3 (mean of independent measurements). **f** GTPase activity of 0.5 μM *Syn*DLP (black) or *Syn*DLP_{HPRN-AAAA} (red). Mean of independent experiments (*n* = 3) and error bars (S.D.) are shown. Data points were fitted using the Michaelis-Menten equation (Eq. (2)). The fit curves are displayed as lines.

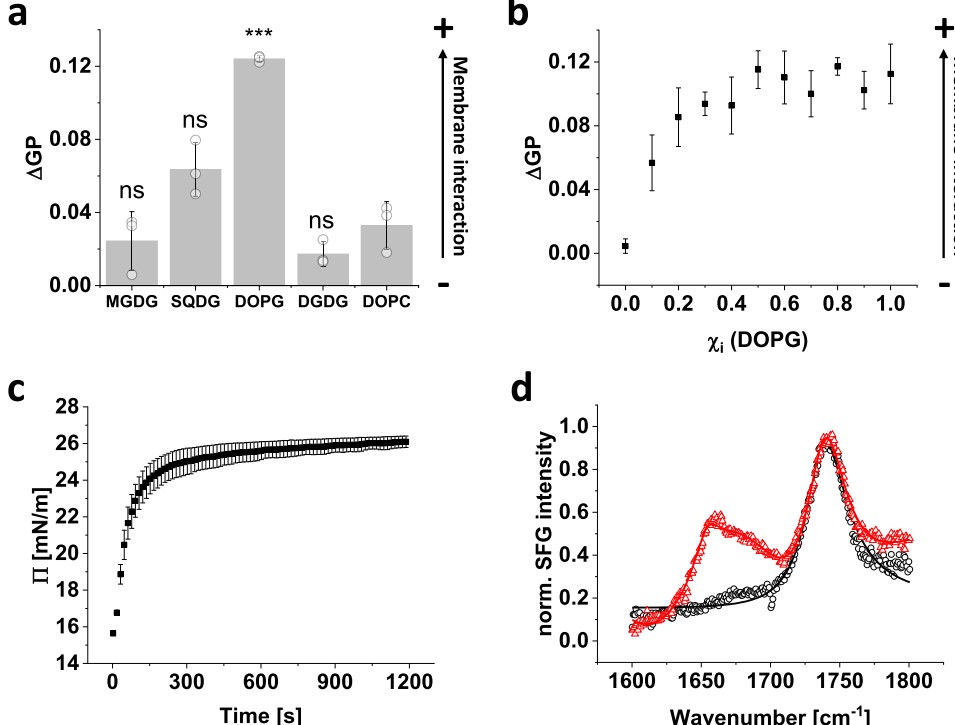

**Fig. 5 | Membrane interaction of *Syn*DLP. a** Fluorescence spectroscopy using Laurdan as a fluorescent probe. LUVs were prepared with 50% DOPC and 50% of the indicated TM lipid (*w/w*) mixed with Laurdan at a 1:500 molar ratio. Laurdan fluorescence emission was measured after 30 min incubation of *Syn*DLP and LUVs and the ΔGP value was calculated from the spectra. Mean of three independent experiments (single measurements shown as circles) and error bars (S.D.) are shown. ns = not significant (*P* > 0.05), *\*P* < 0.05, *\*\*P* < 0.01, *\*\*\*P* < 0.001 based on a two-sided unpaired Student's t-test. The ΔGP values of MGDG (*P* = 0.52), SQDG (*P* = 0.055), DOPG (*P* = 0.00028) or DGDG (*P* = 0.14) are compared to DOPC. Arrow indicates increasing membrane interaction. **b** LUVs were prepared with different DOPG/DOPC molar ratios (Laurdan added at a 1:500 molar ratio). Laurdan fluorescence spectra were recorded after 30 min and ΔGP values (black) were determined from the spectra. Mean of independent measurements (*n* = 4) and error bars (S.D.) are shown. Arrow indicates increasing membrane interaction. **c** Progression of the surface pressure (black) measured over time in parallel to the SFG intensity shown in (**d**). SFG spectroscopy of 0.5 μM *Syn*DLP was carried out on a DMPG monolayer under reaction buffer conditions (without DTT). *t* = 0 s corresponds to the moment of *Syn*DLP addition. Mean of three independent measurements and error bars (S.D.) are shown. **d** The SFG spectrum in the amide I region of the DMPG monolayer (black) and after *Syn*DLP addition and equilibration (red) is shown. A fit of the amide I band is displayed as a line. The mean of three independent measurements is shown. SFG intensity was normalized (0–1) for better comparison.

FTSA curve of *Syn*DLP$_{HPRN-AAAA}$ (Fig. 4e, red) indicated increased hydrophobic, ANS-accessible surface regions, visible as an increased starting F$_{470\ nm}$ value at 20 °C. Most likely, surface regions, which are not covered in the assembly-defective mutant protein, are now ANS accessible, leading to an increased initial ANS fluorescence emission.

Next, the GTPase activity of *Syn*DLP$_{HPRN-AAAA}$ was measured to evaluate the impact *Syn*DLP oligomerization has on its GTP hydrolyzing activity. If *Syn*DLP$_{HPRN-AAAA}$ dimerization via oligomeric interface 2 is assumed (Supplementary Fig. 7), the assembly-defective protein is not expected to form the intermolecular GD contacts anymore that were observed in the *Syn*DLP wt oligomers. The activity of *Syn*DLP wt (black) and *Syn*DLP$_{HPRN-AAAA}$ (red) at different substrate concentrations is shown in Fig. 4f. As the wt, the activity of *Syn*DLP$_{HPRN-AAAA}$ followed a typical Michaelis-Menten kinetic. The fit with a hyperbolic curve revealed a k$_{cat}$ of 17.0 ± 0.2 min$^{-1}$ for *Syn*DLP$_{HPRN-AAAA}$ (Table 3), and thus, the turnover rate was reduced by ~60% in case of the assembly-defective protein. The GTP binding affinity of the mutant appeared not to be significantly affected as indicated by a Mant-GTP binding assay (Supplementary Fig. 12).

### *Syn*DLP interacts with negatively charged TM lipids

While *Syn*DLP is per se a soluble protein, membrane interaction of *Syn*DLP is a prerequisite for catalyzing membrane dynamics. Thus, the membrane interaction propensity of *Syn*DLP was next investigated. First, the membrane interaction of *Syn*DLP with the main TM lipid species MGDG, SQDG, DGDG and PG was tested via fluorescence spectroscopy using the fluorescent dye Laurdan as a probe. Importantly, *Syn*DLP only very weakly interacted with pure (net uncharged) DOPC membranes, which were used as a control (Fig. 5a).

When *Syn*DLP was incubated with large unilamellar vesicles (LUVs) containing 50% of DOPC plus 50% of a respective TM lipid, only little membrane interaction was observed in MGDG- or DGDG-containing membranes, whereas a significant increase of the ΔGP was observed when membranes contained SQDG or DOPG (Fig. 5a). As these two lipids carry negatively charged headgroups, *Syn*DLP-membrane interaction appears to contain a strong electrostatic contribution. Consequently, when *Syn*DLP binding to DOPG-containing membranes was analyzed using DOPC membranes with increasing DOPG mole fractions, an increasing interaction of *Syn*DLP with membranes was observed (Fig. 5b). Based on these observations, all following experiments investigating the interaction of *Syn*DLP with membranes were always conducted with PG-containing membranes.

In the here solved *Syn*DLP structure, we recognized two putative MIDs at the tip of the stalk based either on the quaternary (Fig. 1h, Supplementary Fig. 14a) or the tertiary structure (Supplementary Fig. 14b). However, when these regions were mutated, the isolated recombinant proteins still interacted with DOPG containing LUVs similar to *Syn*DLP wt (Supplementary Fig. 14c+d), and thus these regions (alone) are not responsible for membrane binding of *Syn*DLP. Potentially, either other regions of the protein or a larger area involving multiple *Syn*DLP parts are responsible for membrane interaction.

To obtain structural information about *Syn*DLPs behavior at the lipid-buffer interfaces, we next used sum frequency generation (SFG) spectroscopy to probe *Syn*DLP while binding a DMPG lipid monolayer. An SFG spectrum provides information about the interfacial folding and orientation of proteins. The selection rules of SFG dictate that any signal is exclusively generated by proteins bound to the interface while molecules in solution will not contribute[57]. Measurements were done in a Langmuir trough at a DMPG monolayer, in which the surface pressure was simultaneously monitored[58]. After spreading a DMPG monolayer onto the reaction buffer, an SFG spectrum was recorded at a constant surface pressure of 15 mN/m (Fig. 5c, d). Yet, the surface pressure significantly increased from 15 to ~26 mN/m upon *Syn*DLP addition (Fig. 5c), indicating *Syn*DLP binding to and intercalating into the membrane, resulting in lipid reassembly. When the pure DMPG layer was analyzed via SFG, a resonance signal in the amid I region at around 1738 cm$^{-1}$ arose due to characteristic carbonyl stretch vibrations of the lipid headgroups (Fig. 5d, black). After injection of *Syn*DLP into the subphase of the trough, a broad amide I band (~1625 to ~1700 cm$^{-1}$) was observed, originating from protein backbone vibrations (Fig. 5d, red), which indicated highly ordered binding of *Syn*DLP at the lipid monolayer[57]. The data were fitted using methods outlined previously[59]. The spectra were dominated by a feature at 1654 cm$^{-1}$, which is assigned to α-helical structures. Significantly weaker side bands centered at 1633 and 1680 cm$^{-1}$ are assigned to β-strands structure. The dominance of α-helical SFG signal demonstrated that the structure observed via CD and cryo-EM for the solution state is maintained when *Syn*DLP binds to DMPG lipid interfaces.

Interestingly, the resonance near 1738 cm$^{-1}$ related to the PG carbonyl group remained largely unchanged after *Syn*DLP binding, indicating the lipid head groups remained ordered when interacting with the protein. This supported the assumption of charge interactions involved in the binding mechanism, since such interactions will align the lipid headgroups in the process. To investigate the interactions of *Syn*DLP with the lipid acyl chains, we recorded SFG spectra in the C−D stretching range using lipids with perdeuterated acyl chains. The deuteration allows monitoring the state of the lipid layer without interference by protein C−H modes. The spectra (Supplementary Fig. 15) showed a signal increase for the acyl CD$_3$ modes, which strongly suggested the acyl chains became more ordered with *Syn*DLP binding, which could be the result of the ordering effect of charge-charge interactions between protein and lipid interface or intercalation of protein side chains into the lipid layer.

To glean information about the assembly process of *Syn*DLP on lipid surfaces, we have recorded amide spectra as a function of time (Supplementary Fig. 16). While the spectral shape of the protein amide I band centered around 1653 cm$^{-1}$ was very similar for the interaction time, the intensity grew and then remained constant after 1500 s. The surface tension data showed that binding began to saturate after ca. 250 s. The SFG amplitude, which is sensitive to both the number of proteins and also the orientational order, increased on a similar timescale. While protein assembly often takes place in two steps−fast binding followed by a slower assembly process, the observation that the SFG signal was not lacking behind the surface tension kinetics indicated that *Syn*DLP swiftly formed an ordered layer after lipid binding. The secondary and tertiary structures remained stable throughout the assembly process since the spectral shape of the amide I band did not change significantly. This indicated that *Syn*DLP binds as a stable, folded unit which then quickly assembled into its final binding pose.

As lipid-stimulation of the GTPase activity is a characteristic feature of many (eukaryotic) DLPs[18], we investigated the interplay between *Syn*DLP, PG lipids and GTP. SFG spectra of *Syn*DLP bound to a DMPG monolayer with GTP present in solution (Supplementary Fig. 17) showed that, while the overall amide I signal was somewhat lower than what has been observed without GTP, the spectral shape remained

unchanged. Ostensibly, the conformation of *Syn*DLP did not respond strongly to the presence of GTP. We then determined the *Syn*DLP GTPase activity in the presence of DOPG LUVs (Fig. 6). Yet, unlike many other DLPs, the GTPase activity of *Syn*DLP was not stimulated by membrane binding. As the inclusion of 5 mM MgCl$_2$ in the GTPase assay might lead to clustering of the lipid head groups and, thus, prevent lipid-stimulated GTPase activity, a measurement was additionally performed in the presence of only 0.5 mM MgCl$_2$ ± DOPG LUVs (Supplementary Fig. 11d). Under these conditions, the GTPase activity in the presence of liposomes was slightly increased. Yet, the extent is not comparable to lipid-stimulated GTPase activity observed, e.g., for dynamin[30].

## *Syn*DLP remodels membranes in vitro

Finally, we tested a putative membrane remodeling activity of *Syn*DLP. While, in several cases, the formation of membrane tubes has been observed upon the addition of a DLP to lipids, we did not yet succeed in observing the formation of such structures. However, TM-mimicking LUVs were incubated with *Syn*DLP and their size distribution was analyzed via dynamic light scattering (DLS) and compared to protein and LUVs alone (Fig. 7a). The LUVs were extruded to 100 nm and showed a signal at the expected size (Fig. 7a, black). *Syn*DLP alone showed two broad peaks suggesting fractions of smaller and larger assemblies (Fig. 7a, red). Yet, the mixture of protein and LUVs showed a prominent peak at higher sizes, indicating the formation of protein-LUV-complexes with fused and/or clustered liposomes. Based on this information, *Syn*DLP-triggered membrane fusion was next analyzed with a FRET-based assay using fluorescently labeled MGDG/DOPG LUVs. Fusion of labeled and unlabeled LUVs resulted in an increasing donor fluorescence, since the FRET donor and acceptor are diluted. As simple liposome tethering is not expected to significantly alter the mean distance between the FRET donor and acceptor, this measurement allowed monitoring the kinetics of membrane fusion events. IM30 of *Synechocystis* sp. PCC 6803, a protein with a pronounced fusogenic activity[60], was used as a positive control (Supplementary Fig. 18). When isolated *Syn*DLP wt was mixed with labeled LUVs, the donor fluorescence increased over time, and the increase in donor fluorescence strongly depended on the *Syn*DLP concentration (Fig. 7b, c). Conclusively, we observed a potential fusion activity of *Syn*DLP, as has been observed previously also for other BDLPs[42,43,61]. The two investigated mutants *Syn*DLP$_{C777A}$ and *Syn*DLP$_{HPRN-AAAA}$ also showed an increased donor fluorescence, yet in case of the assembly-defect *Syn*DLP$_{HPRN-AAAA}$ mutant with a lower efficiency (Supplementary Fig. 19). Interestingly, the addition of GTP had no impact on the

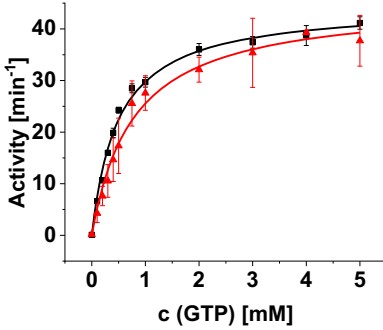

**Fig. 6 | Lipid-stimulation of the *Syn*DLP GTPase activity.** *Syn*DLP GTPase activities in the presence and absence of LUVs were determined in a continuous, regenerative, coupled assay. Comparison of *Syn*DLP under standard measurement conditions (black) and with 50 μM DOPG LUVs (dissolved in reaction buffer, extrusion to 100 nm) added with the assay components (red). Mean of independent experiments ($n = 3$) and error bars (S.D.) are shown. Data points were fitted using the Michaelis-Menten equation (Eq. (2)). Fit curves are shown as lines.

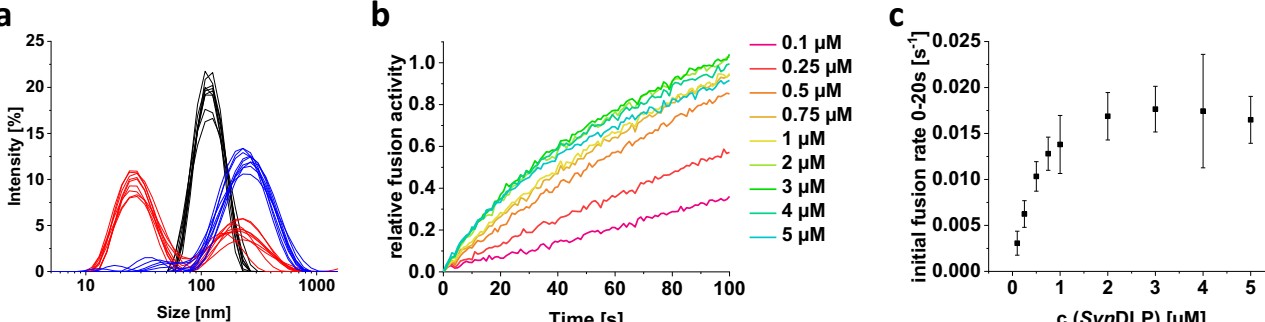

**Fig. 7 | Membrane fusion activity of *Syn*DLP. a** Size distribution of protein and LUVs was analyzed using DLS and relative intensities of the respective sizes are shown. MGDG/DOPG (60%/40%, *w/w*) LUVs (black), *Syn*DLP (red) or a mixture of LUVs and *Syn*DLP (blue) were incubated in reaction buffer and measured. From each condition the size distribution of three independent measurements (+ three technical replicates of each measurement) are shown. **b** Fusion of MGDG/DOPG (60%/40%, *w/w*) LUVs was measured in the presence of increasing *Syn*DLP concentrations using a FRET-based fusion assay. Curves showing the donor fluorescence over the first 100 s after mixing LUVs with protein. The relative fusion activities were calculated as described in the methods section. The curves represent the mean of three independent measurements. The whole measurement as well as the positive control containing 2 μM of the fusogenic protein IM30 are shown in Supplementary Fig. 18. **c** Initial fusion rates (black) were calculated from the curves in (**b**) defined as the slope of a linear regression of the first 20 s. Mean of independent measurements (*n* = 3) and error bars (S.D.) are shown.

fusogenic activity of *Syn*DLP wt (Supplementary Fig. 20). Noteworthy, the here observed membrane fusion activity of *Syn*DLP might, in the end, indicate a membrane destabilizing activity, which results in liposome fusion in vitro. The exact membrane activity of *Syn*DLP will be analyzed with complementing methods in future experiments.

## Discussion

Biogenesis, maintenance and dynamic rearrangement of cyanobacterial TMs are still not understood on the molecular level. In recent years, some proteins have been identified as being involved in the formation of highly curved membrane regions, in membrane fusion and/or membrane repair. One of the currently best-characterized proteins is IM30 (also known as VIPP1), a bacterial homolog of the ESCRT-III core subunit of the eukaryotic ESCRT complex[62–64]. In recent years, several homologs of previously assumed typical eukaryotic proteins involved in membrane dynamics have been identified in bacteria, indicating that processes, which were believed to be of eukaryotic origin, have, in fact, evolved in prokaryotes. Yet, these proteins might fulfill different functions in pro- *vs.* eukaryotes[65].

Cyanobacteria contain several putative genes encoding DLPs, which can be grouped into different clades depending on their genetic context. E.g., in the KGK clade of cyanobacterial DLPs typically a protein containing a KGK domain is encoded downstream of the DLP[44]. *Syn*DLP, a member of this clade, has recently been identified as a cyanobacterial DLP, and the involvement of *Syn*DLP in TM dynamics has been suggested[44]. Isolated *Syn*DLP forms oligomers of ~40–50 molecules in solution in the absence of nucleotides and/or membranes (Fig. 1a, b). Nucleotide- and membrane-independent oligomerization has been reported for eukaryotic DLPs, such as dynamin[10]. However, this is a unique feature not described in the field of bacterial DLPs thus far[35,39,41–43]. Typically, DLPs oligomerize upon nucleotide-binding or upon binding to membrane surfaces[18], and EM micrographs of the here observed *Syn*DLP oligomers (Fig. 1b) looked, in fact, very similar to lipid-free oligomers formed by human Drp1 in the presence of GTP[50]. However, in contrast to Drp1, *Syn*DLP oligomers assemble already in the absence of nucleotides. Within the *Syn*DLP oligomer, individual monomers interact via ionic, polar and hydrophobic interactions, similar to what has been observed in other DLP assemblies[27,50,51]. Based on the *Syn*DLP oligomer structure, defined oligomerization interfaces can be assigned in the stalk region (Supplementary Fig. 7). These include interface 2, which mediates the formation of a stable dimeric unit, and interfaces 1 and 3, both critical for assembly of higher-order oligomers[27,51]. Noteworthy, in contrast to other BDLP structures, such

as the structure of *Np*BDLP that is most closely related to *fusion DLPs*[35,66], the here presented *Syn*DLP oligomer structure resembles a *fission DLP* structure, similar to classical eukaryotic dynamin or Drp1[27,50], that has not been observed in bacteria before. Detailed structural comparisons between *Syn*DLP and bacterial and eukaryotic DLPs demonstrate a close relationship between *Syn*DLP and eukaryotic representatives, indicating *Syn*DLP being the closest known bacterial ancestor of eukaryotic dynamin, Drps and Mx proteins (Fig. 2c, Supplementary Figs. 8, 10).

An intramolecular disulfide bridge, which stabilizes the BSE domain, is observed in the *Syn*DLP structure (Fig. 3a). When the formation of the disulfide bridge was eliminated by replacing C777 with alanine, the resulting protein *Syn*DLP$_{C777A}$ was correctly folded and still formed oligomers, yet, its thermodynamic stability, as well as GTPase activity, were significantly reduced (Fig. 3c–g, Table 2, Table 3), albeit the mutation does not directly affect the active site. This indicated an impact of the BSE domain on the *Syn*DLP GTPase activity. Stabilization of the BSE helix bundle via the formation of an intramolecular disulfide bridge is a concept that has not been described before, and thus far an intramolecular disulfide bridge has solely been observed in the membrane-interacting domain (paddle domain) of the eukaryotic DLP *Ct*Mgm1[67]. Interestingly, the two cysteine residues involved in disulfide bridge formation in *Syn*DLP are conserved across the cyanobacterial KGK clade DLPs (Supplementary Fig. 21). Thus, it can be assumed that a disulfide bridge-stabilized BSE domain is a general feature observable in this clade of cyanobacterial DLPs. In fact, several proteins in the cyanobacterial cytoplasm contain disulfide bridges, and the (in part reversible) formation of disulfide bridges is mediated by the thioredoxin system[68–70]. Thus, it might even be possible that a reversible formation of the disulfide bridge in the BSE domain is involved in the regulation of the *Syn*DLP activity. An extended function of the BSE domain in bacterial DLPs, in general, might be indicated by the observation that all thus far resolved BDLP structures, including the *Syn*DLP structure described here (Fig. 1h), show a BSE domain consisting of a four-helix bundle[35,39,41,43], whereas eukaryotic DLPs typically have a three-helix bundle BSE. However, albeit *Syn*DLP has a four-helical BSE domain, structural alignments indicated a closer relationship to eukaryotic BSE domains (Supplementary Fig. 10), again underlining *Syn*DLPs role as the closest known bacterial ancestor of eukaryotic DLPs.

For most eukaryotic DLPs, such as Dyn1, Drp1, OPA1, Vps1p, Sey1p or Mgm1p, and for BDLPs, like *Np*BDLP or *Ms*IniA, basal GTPase activities were characterized by k$_{cat}$ values in the range of 0.04–5 min$^{-1}$

(reviewed here[21]), which is lower than the here observed SynDLP turnover rate of ~45 min$^{-1}$ (Table 3). However, the GTPase activity of DLPs is often linked to their oligomeric state, and the turnover rate of many DLPs substantially increases upon oligomerization, for example, on an appropriate lipid template[18,21]. Mechanistically, a helical DLP polymer forms on the membrane via defined oligomerization contacts in the stalk, followed by the dimerization of GDs from adjacent rungs. This transverse GD-GD interface leads to the stabilization of critical GD residues and finally assembly-stimulated GTPase activity[22,71,72]. As mentioned earlier, SynDLP forms stable oligomers already in the absence of nucleotides and/or membranes (Fig. 1a, b), and putatively oligomerization stimulates its GTPase activity resulting in the measured high basal GTPase activity. Thus, it is reasonable to assume that GD interactions within the SynDLP oligomer stabilize the enzymatic GTPase core, as observed for other DLPs[22,33,73]. Yet, the determined cryo-EM structure of SynDLP did not indicate transverse contacts between GDs of adjacent monomers. Thus, longitudinal GD contacts within the oligomer might stimulate the SynDLPs GTPase activity. The structure of the SynDLP oligomers revealed only a small interface between adjacent GDs, whereas a large area of the GD of monomer 1 contacts the BSE1 domain of monomer 2 (Supplementary Figs. 8b, 22). The GD-BSE1 contact area is significantly expanded when compared to similar DLP assemblies[27,50] (Supplementary Fig. 8b). Thus, we next analyzed the impact of these contacts on the SynDLPs GTPase activity. A straight-forward mutational approach by substituting the respective contact residues in the GD was not possible, as the residues could not be unequivocally identified in the oligomer structure due to the intermediate resolution of the GD. While we created a mutant where eleven residues in the BSE1 (R12, N16, E20, R23, P26, S30, D33, S35, E38, G42, L45) that contact the GD were replaced, the resulting protein was prone to aggregation and not suitable for subsequent analyses. Therefore, we aimed to analyze an assembly-impaired SynDLP variant (SynDLP$_{HPRN-AAAA}$) to reduce the formation of longitudinal GD-GD and BSE1-GD interactions. Isolated SynDLP$_{HPRN-AAAA}$ showed lowered thermodynamic stability (Fig. 4d, e, Table 2) and was predominantly dimeric (Fig. 4c). Thus, the mutant protein likely did not establish any lateral contacts to adjacent monomers anymore that involve the GD, which potentially increases the SynDLP GTPase activity. The turnover rate of the mutant protein was reduced by ~60% compared to SynDLP wt (Fig. 4f, Table 3). Thus, we propose that uncommon longitudinal interactions between GDs and interactions between the GD and the BSE1 established in the SynDLP oligomers result in the observed high basal GTPase activity. This assumption is further supported by the observation that the disulfide bridge stabilizes the BSE domain, which affects the GTPase activity as described above. Taken together, while intramolecular GD-BSE1 interactions are described for other DLPs[27,52], the SynDLP structure and the analysis of an assembly-defective mutant indicate an additional role of the BSE domain for GTPase activation. It remains to be shown how SynDLP oligomerization and futile GTP hydrolysis might be prevented in vivo, likely by accessory proteins.

As DLPs are generally membrane-active, SynDLP was proposed to be a peripherally membrane-attached protein. SynDLP specifically interacts with negatively charged TM lipids (Fig. 5). Interaction with lipids containing a negatively charged headgroup is well-described for other DLPs, such as eukaryotic Drp1 and dynamin, as well as bacterial BsDynA and MsIniA[29,41,74,75]. A common feature of many DLPs is their ability to tubulate liposomes in vitro, which illustrates their membrane remodeling capabilities[35,43,76,77]. Yet, we did not succeed to observe SynDLP-induced liposome tubulation under the tested experimental conditions. However, SynDLP appeared to be capable of fusing liposomes in vitro in a nucleotide-independent process (Fig. 7, Supplementary Fig. 18), albeit the SynDLP oligomer structure resembles structural elements typically observed in fission DLP structures, as discussed above. The molecular mechanism of DLP-mediated membrane activity still is largely enigmatic. Yet, SynDLP interaction with liposomes may result in membrane destabilization, a process needed for both membrane fusion as well as fission. A membrane destabilizing activity is well conceivable due to the observation that SynDLP not only binds to but intercalates into PG containing membranes (Fig. 5c), which induces perturbations in the lipid structure. Such a disruption of the lipid bilayer structural integrity is also proposed to play a role in atlastin-mediated membrane fusion caused by a C-terminal amphipathic helix[78]. Like SynDLP, BsDynA was shown to fuse membranes in absence of nucleotides, at least in vitro, and also in case of BsDynA, liposome tubulation has not been observed thus far[42]. Recently, an involvement of BsDynA in membrane repair after phage infection has been suggested[36], and it appears reasonable to assume an involvement of SynDLP in membrane repair processes caused by phage infection or other environmental stresses acting on membranes. However, phages infecting Synechocystis sp. PCC 6803 cells are not identified yet, and thus, the exact physiological activity of SynDLP has to be analyzed in future in vivo studies. Potentially, SynDLP is involved in the repair of membranes damaged due to the photosynthetic light reaction, e.g., via vesicle patching or budding.

Also, the exact impact of GTP hydrolysis on the physiological SynDLP function currently remains open. Looking at the structure of a SynDLP monomer (Fig. 1h), the connection between stalk and BSE (hinge 1) seems to be rather rigid, whereas hinge 2 consists of flexible loops with conserved proline residues (P47 and P441) known to facilitate a rotation of the BSE towards the GD[52,79]. The presented SynDLP structure in the apo state shows an 'open' BSE conformation. A 'closed' BSE conformation in SynDLP, which potentially is an intermediate in a GTP hydrolysis cycle, might be observable in structures with bound ligands. The conversion of the 'open' to the 'closed' conformation is thought to act as a power stroke in membrane remodeling processes[18]. Interestingly, the in vitro membrane activity of SynDLP appeared to be nucleotide-independent (Fig. 7, Supplementary Fig. 20).

Taken together, the recently predicted cyanobacterial DLP of Synechocystis sp. PCC 6803 (SynDLP) is a bona fide member of the dynamin superfamily as classified by activity and structure. SynDLP is an active GTPase that forms oligomers in solution. The SynDLP cryo-EM structure revealed folding and oligomerization interfaces known from several eukaryotic fission DLPs in a BDLP and, thus, SynDLP potentially represents a bacterial ancestor of eukaryotic DLPs. Furthermore, unique DLP features, such as an intramolecular disulfide bridge in the BSE domain that affected the thermodynamic stability plus the GTPase activity of SynDLP, and an expanded intermolecular GD-BSE interface were identified in the cryo-EM structure. The presence of such GD interfaces in SynDLP oligomers illustrates a distinctive concept for regulating the basal GTPase activity and would, thus, indicate a so far unique role of the BSE domain in a DLP. SynDLP interacted with negatively charged TM lipids and intercalated into a DMPG monolayer. The classification of SynDLP into fusion vs. fission DLPs is of higher complexity as it showed a membrane destabilizing activity, resulting in liposome fusion in vitro, albeit the structure of the SynDLP oligomer resembles elements typically observed in fission DLPs. Future studies will address the exact mechanism of membrane binding and remodeling including the identification of the membrane binding site, the role of nucleotide binding and hydrolysis, and the in vivo function of SynDLP in the cyanobacterium.

## Methods

### Cloning, expression, and purification of SynDLP

The gene coding for SynDLP (slr0869) was amplified via PCR using genomic Synechocystis sp. PCC 6803 DNA as a template (for primers, see Supplementary Table 1). The NsiI/XhoI restriction digested DNA fragment was ligated into the NsiI/XhoI restriction digested vector pET303-CT/His (Thermo Fisher Scientific, Waltham, MA, USA), resulting in the plasmid pET303-slr0869-CT/His. The mutation in

$Syn$DLP$_{C777A}$ and $Syn$DLP$_{K61A}$ was introduced via site-directed mutagenesis (for primers, see Supplementary Table 1). For expression of $Syn$DLP$_{HPRN-AAAA}$, $Syn$DLP$_{648-665GS}$ and $Syn$DLP$_{667-675GS}$ the pET303-$slr0869$-CT/His plasmid was mutated via Gibson assembly[80] (for primers, see Supplementary Table 1). For protein production, individual clones of transformed $E. coli$ Rosetta-gami™ 2 (DE3) cells (Novagen, Darmstadt, Germany) were used to inoculate 1 L LB medium (100 μg/ml ampicillin), and cells were grown at 37 °C until an OD$_{600}$ of 0.6–0.8 was reached. Expression of $Syn$DLP was induced via the addition of IPTG (1 mM). Cells were grown at 20 °C overnight, harvested by centrifugation the following day, and the resulting pellet was snap frozen in liquid nitrogen and stored at −20 °C until used. The pellet was resuspended in buffer (50 mM NaH$_2$PO$_4$, 300 mM NaCl, 10% glycerol, 10 mM imidazole, pH 8.0) and cells were homogenized using a Potter-Elvehjem device and lyzed with a LM20 microfluidizer® (Microfluidics international cooperation, Westwood, MA, USA) for four rounds at a pressure of 18000 psi. The crude cell extract was centrifuged (15,000 × $g$, 10 min, 4 °C) and the supernatant was mixed with a Ni-NTA matrix (Protino®, Macherey-Nagel, Düren, Germany) equilibrated with 50 mM NaH$_2$PO$_4$, 300 mM NaCl, 10% glycerol, 10 mM imidazole, pH 8.0. After incubation at 4 °C for 2 h, the matrix was washed six times with the same buffer supplemented with either 20, 40, or 50 mM imidazole, respectively. $Syn$DLP was finally eluted with a buffer containing 500 mM imidazole. Next, $Syn$DLP was incubated with 20 mM DTT for half an hour at 4 °C and the protein was further purified via size exclusion chromatography using a self-packed Sephacryl S-400 HR column (Cytiva, Freiburg, Germany) equilibrated with 20 mM HEPES pH 7.4, 0.2 mM DTT on an ÄKTA purifier 10 system (GE Healthcare, Munich, Germany). Main peak fractions were collected and concentrated. Protein concentrations were determined using a reducing agent compatible BCA assay kit (Pierce™, Thermo Fisher Scientific, Waltham, MA, USA). The purity of the protein was estimated by SDS-PAGE (calculated molecular mass of $Syn$DLP: 93 kDa) and the protein was typically ≥95% pure. Per liter $E. coli$ culture, 1–2 mg protein was purified. All subsequent experiments were performed with heterologously expressed and purified protein (except the immunoprecipitation experiment in Supplementary Fig. 1, which showed expression of native $Syn$DLP).

### DTT titration
Purified $Syn$DLP was incubated for 15 min at RT with different DTT concentrations (final concentrations 0.1–500 mM DTT). The samples were mixed with SDS sample buffer, incubated at 95 °C for 5 min, and separated by SDS-PAGE (for the uncropped gel, see Source Data file).

### GTPase assay
GTPase activity was measured using a modified version of a continuous, regenerative, coupled GTPase assay[81]. Reaction buffer (final concentrations: 20 mM HEPES pH 7.4, 150 mM NaCl, 7.5 mM KCl, 5 mM MgCl$_2$, 0.2 mM DTT) was mixed with phosphoenolpyruvate (final concentration: 1 mM). 2.33% ($v/v$) PK/LDH (Sigma-Aldrich Chemie GmbH, Taufkirchen, Germany) and NADH (final concentration: 0.6 mM) (Carl Roth GmbH + Co. KG, Karlsruhe, Germany) were added to obtain a master mix. Different concentrations of GTP (Sigma-Aldrich Chemie GmbH, Taufkirchen, Germany) dissolved in 20 mM HEPES pH 7.4 were added to the master mix, and the solutions were incubated at 4 °C for 15 min to convert all remaining GDP to GTP. The protein (0.5 μM final concentration) and pure buffer (blank) were placed into a 96-well plate and mixed with the GTP-containing master mix, resulting in a final volume of 150 μl per well. The absorption at 340 nm was observed with a microplate reader (FLUOstar® Omega, BMG Labtech GmbH, Ortenberg, Germany) using the software Omega (version 1.3) at 30 °C over 2–3 h to make sure that even at the lower GTP concentrations all NADH was oxidized. The activities at different substrate concentrations were calculated using the MARS Data Analysis software

(version 2.10 R3) and MS Excel (version 2208) as follows: The absolute value of the slope of the blank measurement was subtracted from the absolute value of the maximum linear absorption decrease at 340 nm, yielding the corrected decrease of the absorption at 340 nm over time ($\Delta A_{corr}$). The GTP hydrolyzing activity was calculated with Eq. (1):

$$\text{Activity} = \frac{\Delta A_{corr}}{d * \varepsilon * c_{Protein}} \quad (1)$$

$\varepsilon$ refers to the molar extinction coefficient of NADH at 340 nm (6220 M$^{-1}$ cm$^{-1}$), and $d$ is the thickness of 150 μl sample volume in a 96-well plate (0.38 cm). The calculated activities were plotted against the GTP concentrations. The data points were fitted with a Michaelis-Menten equation (Eq. (2)) using Origin™ (version 9.60) (OriginLab Corporation, Northampton, MA, USA) to determine the turnover rate ($k_{cat}$) and the Michaelis-Menten constant ($K_m$).

$$\text{Activity}([\text{GTP}]) = \frac{[\text{GTP}] * k_{cat}}{[\text{GTP}] + K_m} \quad (2)$$

### Analytical size exclusion chromatography
A Superose® 6 Increase 3.2/300 column (Sigma-Aldrich Chemie GmbH, Taufkirchen, Germany), equilibrated with reaction buffer, was loaded with 30 μl of a 10 μM protein solution. Proteins were eluted using an ÄKTA purifier 10 system (GE Healthcare, Munich, Germany) at 7 °C and a flow rate of 0.03 ml/min. The standard proteins thyroglobulin (669 kDa) Ferritin (440 kDa), Aldolase (158 kDa), Conalbumin (75 kDa) and Ovalbumin (44 kDa) (Sigma-Aldrich Chemie GmbH, Taufkirchen, Germany) were used for protein mass estimation. Chromatograms were recorded using the software UNICORN (version 5.10).

### Circular dichroism spectroscopy
The CD spectrum of 1 μM protein was recorded in 10 mM HEPES, pH 7.4, using a JASCO J-1500 CD spectrometer with an MPTC-490S temperature-controlled cell holder (JASCO cooperation, Tokyo, Japan) and the software JASCO Spectra Manager (version 2.15.01). The spectral range was 200–250 nm with a scan rate of 50 nm/min, 1 mm cell length, 1 nm steps, 5 nm bandwidth, 1 s data integration time, 8-time accumulation at 20 °C. For measurements of the thermal stability, the spectra were recorded from 200 to 250 nm with a scan rate of 100 nm/min, 1 mm cell length, 1 nm steps, 5 nm bandwidth, 1 s data integration time, 1-time accumulation at increasing temperatures starting from 20 °C to 84 °C recorded in 2 °C steps and a heating rate of 1 °C/min. Each spectrum was smoothed using the Savitzky-Golay filter from the software JASCO Spectra Analysis (version 2.15.01). Three individual measurements were averaged for each sample. For this purpose, the data sets were interpolated, as the measurements were recorded at slightly different actual temperatures. The interpolated data sets were normalized and averaged, resulting in a melting curve. The transition temperature ($T_m$) was determined using an adapted Boltzmann fit (Eq. (3)). This fit is based on a two-state unfolding mechanism and enables linear slopes in the plateau areas of the curve:

$$\theta_{meas}(T) = \frac{(T * m_N + \theta_N) - (T * m_D + \theta_D)}{1 + e^{\frac{T - T_m}{dT}}} + (T * m_D + \theta_D) \quad (3)$$

$\theta_{meas}$ refers to the measured ellipticity at 222 nm and $T$ is the temperature. $\theta_N$ and $\theta_D$ are the ellipticities at the plateau regions of the native and the denatured protein, $m_N$ and $m_D$ are the slopes of the corresponding plateaus. Data procession and the fit were performed using Origin™ (version 9.60) (OriginLab Corporation, Northampton, MA, USA).

## ANS fluorescence thermal shift assay

Proteins were mixed with the fluorescent dye 8-anilinonaphthalene-1-sulfonic acid (ANS), which shows increased fluorescence emission when bound to hydrophobic pockets found at surfaces of folded proteins or to hydrophobic regions becoming accessible due to protein unfolding. Thus, ANS can be used to monitor protein unfolding. For an FTSA, SynDLP (5 μM) was mixed with 50 μM of the fluorophore ANS in reaction buffer. Fluorescence emission was recorded from 400 to 600 nm using a JASCO FP-8500 fluorescence spectrometer (JASCO cooperation, Tokyo, Japan) and the software JASCO Spectra Manager (version 2.9.0.7) upon excitation at 370 nm. Integration time was 0.1 s, and the excitation and emission slits corresponded to a spectral resolution of 2.5 nm. Spectra were recorded with a scan rate of 200 nm/min, 1 nm steps at temperatures ranging from 20 to 80 °C in 1 °C steps, and a heating rate of 1 °C/min. The fluorescence intensity at 470 nm was used as a measure of the SynDLP folding state. Three independent measurements were combined for each sample. The data sets were interpolated due to different actual temperatures and then averaged to obtain a melting curve of the protein. The transition temperature was determined using an adapted Boltzmann fit similar to Eq. (3) (Eq. (4)):

$$F_{meas}(T) = \frac{(T * m_N + F_N) - (T * m_D + F_D)}{1 + e^{\frac{T - T_m}{dT}}} + (T * m_D + F_D) \qquad (4)$$

$F_{meas}$ denotes the measured fluorescence intensity at 470 nm, while $T$ is the temperature. $F_N$ and $F_D$ are the fluorescence intensities at the plateau regions of the native and the denatured protein, $m_N$ and $m_D$ are the slopes of the corresponding plateaus. It is important to note that fitting using Eq. (4) could not be applied to the entire measured temperature range, as the ANS fluorescence strongly depends on the temperature besides binding to a folded or denatured protein. Therefore, the adjusted data included a temperature range of 20–25 °C that captured the transition phase. Data procession and the fit were performed using Origin™ (version 9.60) (OriginLab Corporation, Northampton, MA, USA).

## Preparation of large unilamellar vesicles

The lipids DOPG (1,2-dioleoyl-*sn*-glycero-3-phosphoglycerol), DOPC (1,2-dioleoyl-*sn*-glycero-3-phosphocholine), DMPG (1,2-dimyristoyl-*sn*-glycero-3-phosphoglycerol), MGDG, DGDG and SQDG as well as the fluorescently labeled lipids LissRhod-PE (Lissamine Rhodamine PE; 1,2-dioleoyl-*sn*-glycero-3-phosphoethanolamine-N-(Lissamine Rhodamine B sulfonyl) (ammonium salt)) and NBD-PE (1,2-distearoyl-*sn*-glycero-3-phosphoethanolamine-N-(7-nitro-2-1,3-benzoxadiazol-4-yl)) were all purchased from Avanti Polar Lipids, Inc. (Birmingham, AL, USA). For the preparation of LUVs, the organic solvent (CHCl₃/MeOH 2:1 (*v/v*)) was evaporated under a gentle stream of nitrogen gas followed by vacuum desiccation overnight to remove any traces of solvent. The dried lipid film was hydrated in an appropriate buffer and LUVs were prepared by five cycles of freezing in liquid nitrogen and thawing at 37 °C. For the membrane fusion assay, the LUVs were extruded 15 times through a 100 nm filter (Nucleopore Track-Etch Membrane, Whatman, Sigma-Aldrich GmbH, Taufkirchen, Germany) using an extruder (Avanti Polar Lipids, Inc., Birmingham, AL, USA).

## Laurdan fluorescence measurements

The fluorescent dye Laurdan (6-dodecanoyl-*N,N*-dimethyl-2-naphthylamine) incorporates into lipid bilayers. Its fluorescence emission characteristics depend on the polarity of the environment. Determining Laurdan fluorescence emission spectra can be used to monitor changes in the membrane polarity, e.g., caused by protein adhesion to a membrane surface. The effect of SynDLP on different model membrane systems containing DOPC and varying amounts of TM lipids (MGDG, DGDG, SQDG, DOPG) was tested. Laurdan (Sigma-Aldrich

Chemie GmbH, Taufkirchen, Germany) was added at a molar ratio of 1:500 to a 0.5 mM lipid solution prior to LUV formation. LUVs were prepared in 20 mM HEPES, pH 7.4 as described above. 0.1 mM LUVs were incubated with 0.5 μM SynDLP in 20 mM HEPES pH 7.4 for 30 min at 25 °C. Laurdan fluorescence emission spectra were measured at 25 °C from 400 to 550 nm using a Fluoromax-4 spectrometer (Horiba Scientific, Kyoto, Japan) and the software FluorEssence (version 3.8) with excitation at 350 nm. The excitation and emission slit widths corresponded to a spectral resolution of 3 nm. The data were further analyzed using MS Excel (version 2208) and Origin™ (version 9.60) (OriginLab Corporation, Northampton, MA, USA). Spectral changes were quantified using the Generalized Polarization (GP) value[82], which was calculated for each spectrum (Eq. (5)):

$$GP = \frac{I_{440} - I_{490}}{I_{440} + I_{490}} \qquad (5)$$

Here, $I_{440}$ and $I_{490}$ are the Laurdan fluorescence emission intensities at 440 and 490 nm, respectively. ΔGP values were obtained by subtraction of the GP value of LUVs without protein.

## SFG spectroscopy

SFG spectroscopy was used to obtain selective information about the orientation and conformation of SynDLP at membrane-buffer interfaces. For a more detailed description of this method concerning interfacial protein structure elucidation, see, for example,[57,83–85].

All SFG experiments were performed according to established procedures[59,86] using SFGTools (https://github.com/james-d-pickering/SFGTools) and Matlab (version 2022a). Briefly, two laser beams (one visible and one tunable laser) overlap at the sample to generate light at their sum-frequency ($\omega_{VIS} + \omega_{IR} = \omega_{SFG}$). Pulses were provided by a Ti:sapphire laser (Mai Tai, Spectra-Physic, Santa Clara, CA, USA), amplified (Spitfire Ace, Santa Clara, CA, USA) and pumped (Nd:YLF laser, Empower, Spectra-Physics, Santa Clara, CA, USA). The resulting beam had a pulse repetition rate of 1 kHz and a duration of about 40 fs. The power was about 5 mJ at 800 nm. One part of the beam was used to generate the IR beam in an optical parametric amplifier (TOPAS, Light Conversion, Vilnius, Lithuania). The SFG signal was detected by an EMCCD camera (Newton, Andor, Belfast, Northern Ireland). All spectra were recorded in the polarization setting ssp (s-SFG, s-VIS, and p-IR) and averaged for 600 s, respectively. For further spectral analysis, the background was subtracted, and the spectra were normalized using a non-resonant reference spectrum (gold-coated silicon wafer). The spectra were fitted assuming a Lorentzian line shape and using the software Origin™ (version 9.60) (OriginLab Corporation, Northampton, MA, USA).

## Dynamic light scattering (DLS)

Size distributions of proteins and LUVs were determined via DLS. MGDG/DOPG (60%/40%, *w/w*) LUVs were prepared in 20 mM HEPES pH 7.4, 0.2 mM DTT and extruded to 100 nm, as described above. Either 1 μM SynDLP, 0.1 mM LUVs or a mixture of both were incubated at RT in reaction buffer for 15 min and measured in a Zetasizer Pro (Malvern Panalytical Ltd, Malvern, UK) using the software ZS Xplorer (Version 2.3.1.4) at 25 °C using backscatter. After an equilibration time of 60 s, three individual samples were measured three times and analyzed via the manufacture's software, finally yielding in intensity-weighted size distribution.

## Membrane fusion assay

Membrane activity of SynDLP was tested using a FRET (Förster resonance energy transfer)-based LUV fusion assay, as described in detail recently[60,87]. Unlabeled MGDG/DOPG (60%/40%, *w/w*) LUVs and labeled LUVs, containing 0.8 mol% of the FRET dyes LissRhod-PE and NBD-PE each, were prepared in 20 mM HEPES pH 7.4, 0.2 mM DTT, as

described. Unlabeled and labeled LUVs were mixed at a ratio of 9:1 ($v/v$), and when the two LUV species fused, the FRET dyes redistributed over the membrane leading to decreased FRET and increased donor emission. Mock fused LUVs, containing solely 1/10 of the fluorescently labeled lipids, were produced and measured to correct for bleaching. Protein solutions were incubated in reaction buffer at 25 °C 10 min prior to the measurement. After that, the solution containing SynDLP in different concentrations (0–5 μM) and the mixture of unlabeled and labeled LUVs (final concentration: 0.1 mM) were rapidly mixed, and the measurement was started immediately. The fluorescence emission of the FRET donor NBD-PE was monitored at 535 nm for 15 min at 25 °C after excitation at 460 nm using a Fluoromax-4 spectrometer (Horiba Scientific, Kyoto, Japan) and the software FluorEssence (version 3.8). The slit widths for excitation and emission corresponded to a spectral resolution of 2 and 10 nm, respectively. The fusion activity of 2 μM of the cyanobacterial IM30 protein, which is well-known to trigger the fusion of membranes containing TM lipids[60,88–90], was measured as a positive control. The data were further analyzed using MS Excel (version 2208) and Origin™ (version 9.60) (OriginLab Corporation, Northampton, MA, USA). Relative fusion activities were calculated via conversion of the raw fluorescence data by Eq. (6):

$$\text{Fusion activity}\,(t) = \frac{I - I_0}{I_M - I_0} \tag{6}$$

Here, $I_0$ is the NBD-PE fluorescence intensity of the negative control measured without protein, $I_M$ is the intensity of the mock fused LUVs and $I$ the measured sample at every point in time $t$. The initial fusion rates were determined by the slope of linear regression of the first 20 s of every fusion curve.

### Negative staining electron microscopy

For negative-staining electron microscopy, 3.5 μl of the sample was applied to glow-discharged (PELCO easiGlow Glow Discharger, Ted Pella Inc., Redding, CA, USA) continuous carbon grids (Cu 300 hex grids; Electron Microscopy Sciences, Hatfield, PA, USA; in-house coated with carbon film). The sample was incubated on the grid for 1 min. Then the grid was side-blotted using filter paper, washed with 3.5 μl water, stained with 3.5 μl 2% uranyl acetate for 30 s, and air-dried. The grids were imaged with a 120 kV Talos L120C electron microscope (Thermo Fisher Scientific, Waltham, MA, USA; FEI Company, Hillsboro, OR, USA) equipped with a CETA camera at a pixel size of 4.05 Å/pixel (36 kx magnification) at a nominal defocus of 1.0–2.5 μm.

### Cryo-electron microscopy (cryo-EM) and image processing

SynDLP samples were prepared by applying 3.5 μl SynDLP (~3.0 mg/mL in 5 mM MgCl₂, 7.5 mM KCl, 20 mM HEPES pH 7.4) to glow-discharged (PELCO easiGlow Glow Discharger, Ted Pella Inc., Redding, CA, USA) Quantifoil grids (R1.2/1.3 Cu 200 mesh; Electron Microscopy Sciences, Hatfield, PA, USA). The grids were plunge-frozen in liquid ethane using a Thermo Fisher Scientific Vitrobot Mark IV (Thermo Fisher Scientific, Waltham, MA, USA) set to 90% humidity at 20 °C (blotting force −10, blotting time 3 s). A total of 8322 micrographs were recorded on a 200 kV Talos Arctica G2 (Thermo Fisher Scientific, Waltham, MA, USA) electron microscope equipped with a Bioquantum K3 (Gatan, Inc., Pleasanton, CA, USA; controlled by Digital Micrograph (version 3.32.2403.0)) detector operated by EPU (version 2.11.0.2368REL) and TIA (version 5.0.0.2896) together with FluCam viewer (version 6.15.3.22415) (Thermo Fisher Scientific, Waltham, MA, USA). Micrographs were collected as 30-frame movies in super-resolution mode at 0.8685 Å/pixel and a cumulative dose of 26.5 e⁻/Å² and a nominal underfocus of 2.0–4.0 μm.

Movie frames were gain-corrected, dose weighted, and aligned in super-resolution using cryoSPARC Live[91] (version 3.2). Initial 2D classes

were produced using the auto picker implemented in cryoSPARC Live. All the following image processing steps were performed using cryoSPARC (version 3.2). The 7 best-looking classes were used as templates for the template picker. The resulting 10,000,000 particles were extracted with 450 px box size (Fourier cropped to 256 px) and subjected to multiple rounds of 2D classification. The cleaned particle set contained 2,685,000 particles. The particles were further divided by ab initio reconstruction into two classes. The first class contained 1,806,000 particles and was used for further processing. The ab initio model of the first class was further refined using multiple rounds of non-uniform refinement and hetero-refinement with imposed C2 symmetry. Finally, the particles were reextracted to the full resolution (super-resolution) and subjected to non-uniform refinement (including group and per-particle CTF refinement), followed by local refinement. The final reconstruction was determined to have a global resolution of 3.7 Å (auto-masked, FSC = 0.143) and was based on 977,200 particles. Local filtering and determination of the local resolution was performed using cryoSPARC.

### Cryo-EM map interpretation and model building

The 3D reconstruction was B-factor sharpened (B = −184.9 Å²) in cryoSPARC. The handedness of the final map was determined by rigid-body fitting the BSE-domain of NpBDLP[35] into the final map using ChimeraX[92] (version 1.2.5). The locally filtered map was used for de novo model building of the BSE-domain in Coot[93] (version 0.95). The resolution in the periphery of the map did not allow model building of the GD. Thus, a model of the GD (aa 48–423) was generated using AlphaFold2[94,95]. To integrate the AlphaFold2 model with the de novo model, an overlap of 40 aa at the N and C-terminus of the GD was included in the model. The AlphaFold2 model was flexibly MDFF fitted to the 3D reconstruction using ISOLDE[96] (version 1.2) and fused to the de novo model. The resulting monomer model includes aa 1–793. The monomer model was manually refined using Coot and ISOLDE before rigid body extending a total of 8 monomers to fill the complete 3D reconstruction. Non-crystallographic symmetry (NCS) parameters for the octamer were obtained by phenix.find_ncs (Phenix version 1.18). The octamer model was subjected to two cycles of auto-refinement with phenix.real_space_refine[97] (with NCS constraints and NCS refinement) and local map sharpening using LocSCALE[98] (version 0.1). After final inspection, the model was validated in phenix.validatiopn_cryoem[99]/ Molprobity[100].

### Reporting summary

Further information on research design is available in the Nature Portfolio Reporting Summary linked to this article.

## Data availability

Source data are provided as a Source Data file. SynDLP cryo-EM map and the corresponding refined atomic model were deposited in the corresponding databank under the following IDs: EMD-14993 and PDB ID-7ZW6 [https://doi.org/10.2210/pdb7ZW6/pdb]. Source data are provided with this paper.

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

## Acknowledgements

This work was funded by the Max-Planck Graduate Center at the Max Planck institutes and the University of Mainz. We would like to thank Nadja Hellmann for advice on data analysis and Carmen Siebenaller for supply of purified IM30 protein. The authors gratefully acknowledge the electron microscopy access time and computing time granted by the biological EM facility of the Ernst-Ruska Centre at the Forschungszentrum Jülich. In this regard, we thank Thomas Heidler and Pia Sundermeyer for maintaining the electron microscopes and Daniel Mann for maintaining the processing computers. The authors gratefully acknowledge the computing time granted through Jülich Aachen Research Alliance (JARA) on the supercomputer JURECA at Forschungszentrum Jülich (https://jlsrf.org/index.php/lsf/article/view/171).

## Author contributions

L.G., R.J., and D.S. designed research. L.G., R.J., and W.E.Z. cloned, expressed, and purified the proteins. L.G., R.J., and W.E.Z. performed biochemical and biophysical analysis of the proteins. B.J. and C.S. prepared cryo-EM samples and determined the cryo-EM structure. J.F., R.J., and T.W. performed and analyzed the SFG spectroscopy measurements. L.G., B.J., R.J., T.W., M.B., C.S., and D.S. prepared the manuscript. All authors have read and agreed to the published version of the manuscript.

## Funding

## Competing interests

The authors declare no competing interests.
