## [Peer Review File · Nature Communications]

SynDLP is a dynamin-like protein of *Synechocystis* sp. PCC 6803 with eukaryotic featuresREVIEWER COMMENTS

Reviewer #1 (Remarks to the Author):

The manuscript by Gewehr and colleagues describes the structural analysis of a cyanobacterial dynamin-like protein. The authors used cryo-EM to solve the oligomeric structure of SynDLP with a 3.7 Å resolution. SynDLP form short oligomeric filaments of around 50 monomers. The oligomerization is stabilized via stalk interfaces, and thus resembles the oligomerization of eukaryotic dynamins. The authors identified a unique cysteine bridge in the BSE domain of the protein. The cysteines involved seem to be conserved in certain cyanobacterial clades. Removal of the disulfide bridge by mutation of the cysteines reduces GTP hydrolysis activity of SynDLP. Although the overall assembly of the SynDLP oligomer resembles that of classical dynamin, the contacts within the adjacent GTPase domains is different. In line with a different assembly, the basal GTPase activity of SynDLP is comparatively high. As shown for other bacterial DLPs before, SynDLP is able to fuse liposomes in vitro, as shown by a FRET-based assay.

This manuscript provides the first oligomeric structure of a full-length bacterial dynamin in solution. Therefore, this work is an important addition to our existing knowledge. The experiments are carefully executed and the data are of high quality. There are only some minor points that should be addressed by the authors in a revision.

The structure was solved in the absence of nucleotide. Did the authors check for absence of nucleotide after purification? Did the authors try to get a cryo structure with added GTP and magnesium?

Line 42: Bacterial dynamins were described before the first structure was reported by Low and Löwe. A bioinformatic study identified bacterial DLPs as early as 1999 (DOI: 10.1016/s0962-8924(98)01490-1)

Lines 71/72: This sentence is redundant with the one in line 42. Again, the bacterial DLPs have been identified (by bioinformatics) before.

Figure S1: What are the other bands – cross reactions from the antibodies?

Line 102: “code” might be replaced by “encode”

Line 126. A brief explanation why the GD domain have higher flexibility would help. Generally, the GD domain is a well-structured domain, but I assume the authors mean that the GTPase domain may undergo rigid body movements and hence the dynamics?

Figure 5: Membrane binding measurements with Laurdan GP is not ideal. I agree that binding of a protein to the membrane changes the GP values, but this is a very indirect measurement. Laurdan GP will also vary in membranes with different lipid compositions. A simple assay would be co-sedimentation, but ideal would be a more quantitative interaction method such as SPR. The manuscript would greatly benefit from a better description how SynDLP binds to the membrane. Based on the structure the membrane binding paddle and the residues involved can likely be identified. The authors should at least elaborate more on the membrane binding of SynDLP.

Line 312: 30 min (instead of 30 in)

Figure 7: Did the authors add nucleotide or magnesium to the fusion assays?

Line 388: Please introduce the term KGK clade. The paper will be read not only by cyanobacterial specialists. Is there something special in the KGK clade that can be related to the disulfide bridge being necessary?

Discussion: Generally, fission and fusion of membrane is mechanistically related and involves a hemifusion state. Therefore, lipid mixing (this is what is shown in Figure 7) is not necessarily a proof of fusion in an in vivo setting. The eukaryotic Dyn2 is also a fission GTPase in vivo, but in vitro it can induce lipid tethering (and mixing) under certain experimental conditions. The discussion should therefore be slightly tuned down.

Reviewer #2 (Remarks to the Author):

Gewehr et al report the cryoEM structure of oligomerized SynDLP, a dynamin-related protein from the cyanobacterium *Synechocystis* sp PCC 6803. Based on their structural analysis, they performed a biochemical study to conclude that SynDLP uses an uncommon activation mechanism for GTP hydrolysis.

The structural work is noteworthy and remarkable since SynDLP appears to be the first bacterial dynamin-like protein that is closely related to eukaryotic dynamin and Drp1, two key molecules of membrane remodeling in higher eukaryotes. However, the structural analyses and representations are not sufficient to understand the architecture and significance of the new model. Also, the biochemical

data set appears incomplete; additional experiments are required to support the conclusions. Some data may not be correctly interpreted.

Major:

1.) This is the first structural data of this type of bacterial dynamin-like protein, but from the current presentation, it is difficult to understand the DynDLP architecture. Structural figures are not properly labelled and are too small to recognize any details. The authors have to put more efforts to explain the newly determined structure to the reader. In particular, Fig. 2 should contain a structure-based domain representation of DynDLP. A topology plots would be helpful to understand the detailed architecture. Fig. 2e should be enlarged and properly labelled (GTPase domain, stalk, BSE, the two hinges including the proline residues, the membrane-binding site, nucleotide binding site, some secondary structure elements, etc). The four-helix architecture of the BSE should be properly displayed. Higher magnifications of functionally relevant sites should be provided, in which key amino acids are properly labelled. Although the resolution of the cryoEM density is relatively low in some parts, the fitting with the AF2 model, as done in this study, should allow the identification of key amino acids in some of the interfaces. Accordingly, molecular details of all assembly interfaces should be provided and again properly labelled (unlike in Supp. Fig. 6). The disulfide bridge needs to be shown in detail and labelled in a magnified view (Fig. 3a is not helpful in this regard).

2.) From what I understand, SynDLP could be the long sought-after bacterial ancestor of eukaryotic dynamin, MxA and Drp1; the other dynamin-related proteins reported so far, show somewhat different assemblies and modes of action. However, to make this important claim, a thorough structural comparison of SynDLP to other reported dynamin proteins from bacteria and eukaryotes has to be performed. rmsds of isolated domains (in particular the stalk and BSE) should be reported, topologies and assembly interfaces should be compared, surface conservation plots should be provided in a separate main figure. At the moment, the available structural data are not fully mined.

3) Figure 5: The membrane-binding site in dynamin-related proteins is normally localized in a loop or domain at the tip of the stalk. Based on the performed analysis, it seems reasonable to assume that it involves a patch of positively charged amino acids in the related SynDLP loop at the tip of the stalk. It will be straightforward to identify and describe the membrane-binding site with a small set of mutations, which would strengthen the impact of the study.

4.) The GTPase domain is the most highly conserved region in dynamin proteins. In fact, essentially all examined dynamin-related and septin-related proteins use a GTPase domain-mediated dimerization mechanism via the 'G-interface' for stimulated GTPase activity. Furthermore, the G-interface is structurally well characterized in various dynamins, but in many cases, it is observed only in the presence of GTP/GDP-AlF₃, not without nucleotide. Given the close structural resemblance of SynDLP

and dynamin, it is reasonable to assume that SynDLP also uses a related dimerization-dependent GTPase mechanism. On the other hand, it has already been described that the assembly of dynamin superfamily proteins via the stalk or GED can indirectly affect GTP hydrolysis rates (see for example, Gao et al, Nature 2010, for related studies on dynamin-like MxA). The 'non-canonical' GTPase domain-BSE contacts shown in Supp. Fig. 4 may have a similar indirect effect on GTPase activation. Thus, the notions that 'non-canonical' contacts in the GTPase domain stimulate GTP hydrolysis (title, abstract, line 216) and the idea of a GTPase-regulating function for the BSE (line 95) are likely misleading. Some further experimental analysis is warranted: How is the GTPase activity affected for mutants in the 'non-canonical' GTPase domain interfaces (rather than relying on mutations far away from the GTPase domain)? Does SynDLP display a protein concentration-dependent GTPase stimulation? Can GTPase activity be abolished by a point mutation in the G-interface? This would unambiguously demonstrate related GTPase mechanisms of dynamin and SynDLP.

5.) GTPase activity: Dynamin family proteins use a dimerization-mediated GTPase mechanism which is based on different assumptions from the classical Michaelis-Menten kinetics. In particular, a lower observed K_m value is not necessarily related to an altered nucleotide binding affinity, but it could also be explained by a different assembly kinetics of the GTPase domains in oligomerized vs. dimeric constructs. How should a mutation in the stalk directly affect the nucleotide-binding affinity in the GTPase domain? A direct comparison of nucleotide-binding affinities, for example using mant-nucleotides, should be performed to make such a claim.

As already stated in the manuscript, the reported 'basal' GTPase activity of SynDLP appears unusually high when compared to other dynamin-related proteins. Were only initial rates considered for the Michaelis-Menten kinetics in the 2-3h measurement period reported (which seems very long for such a fast reaction)? The inclusion of 5 mM $MgCl_2$ could prevent any liposome-stimulated GTPase activity due to clustering of the lipid head groups. GTPase measurements should be repeated with liposomes in the presence of 0.5 mM $MgCl_2$ and 150 mM KCl instead of NaCl since dynamin proteins use a K^+ -dependent GTPase mechanisms (Dyda et al, Nature 2010).

6.) Fig. 7: The structural similarity of SynDLP and dynamin/Drp1 points to a related mechanism of the proteins in membrane constriction. From this point of view, the membrane fusion properties of SynDLP reported in Fig. 7 appear highly surprising. However, the assay on its own seems not sufficient to support the conclusions of SynDLP being a fusogen. The undefined oligomeric assemblies of SynDLP observed in solution may just induce non-specific liposome aggregation, which may also explain the fluorescent increase in the applied assay. The authors should complement their experiments with negative-stain EM analyses of SynDLP and liposomes in the absence and presence of GTP (again using maximally 0.5 mM $MgCl_2$) to observe putative fusion or fission states. Alternatively, a light microscopy-based liposome tethering assay could be employed, similar as shown for IniA in Wong et al, Nat Comm 2019 – in fact, this article also contains nicely labelled structural figures for comparison.

8.) A detailed description of the cellular function of SynDLP may be beyond the scope of the manuscript. However, since the SynDLP ko strain has already been generated (Supp. Fig. 1), it would be neat to at least describe whether the thylakoid membranes appear altered in this mutant or whether photosynthesis is possibly affected? To suggest putative roles of SynDLP in phage infection or membrane repair (line 441-447) seems too speculative without any supporting experimental data.

Minor:

Description of the results could be shortened at several positions. For example, line 105-109 could go to the Methods, Fig.1a could go to the supplement (ideally with some further information on the protein purification), Fig. 1b could be combined with Fig. 2. The reduced melting temperature of the C777A mutant is maybe not a huge surprise so this paragraph can be shortened (line 149-167). Also the ANS assay and the fitting procedures of the curve do not to be explained in depth in the results (line 169-179) and the description of the biochemical characterization of the interface-3 mutant (line 229-249) could be shortened. This would leave more space to elaborate on the structural analysis.

Fig. 2a: What is the radius that the SynDLP oligomer would embrace? It seems rather a low curvature radius compared to dynamin?

CryoEM analysis:

Is there any chance to improve the resolution of the GTPase domain by local refinement and masking (as for example, in Chaaban and Carter, Nature 2022)?

Fig. 5: SFG spectroscopy appears to be a rather specialized method for monitoring protein-membrane interactions. For example, what do we learn from Fig. 5d? Does this analysis provide useful information to the current manuscript?

Typos

Line 140:

Supp. Fig. 2b does not exist.

Reviewer #3 (Remarks to the Author):

Major comments:

Nucleotide State

1. "...Yet, in contrast to other BDLPs, SynDLP forms high-molecular mass oligomers already in complete absence of lipids and/or nucleotides..."

It is unclear how the authors conclude that lipids and/or nucleotides are "completely" absent, since nucleotides or native lipids might have co-purified with the protein. With a local resolution distribution between 3.7 Å to 7 Å, it would be challenging to detect lipid density in certain areas of the structure. Do the authors have other evidence to support this statement? Have the authors checked for lipids or nucleotides by mass spectrometry or other means?

2. In the discussion, "...It remains to be shown how SynDLP oligomerization and futile GTP hydrolysis might be prevented in vivo, likely by accessory proteins..."

The authors assume here that the GTP hydrolysis in vivo is futile. This reviewer has concerns about the GTPase assay setup. A negative control of protein without nucleotide is lacking in the final figures. In addition, unlike all other experiments, the authors conduct the GTPase activity assay in the presence of salt (150 mM NaCl, 7.5 mM NaCl). Given the sensitivity of DLP self-assembly to changes in salt conditions, it would be interesting to know if the authors observed any differences in GTPase activity or SynDLP oligomerization under the same salt conditions used for the other experiments (for cryo-EM: 5 mM MgCl₂, 7.5 mM KCl; for Laurdan fluorescence, CD and purification: no salt).

Oligomerization State

Related to the nucleotide comments above, the final buffer for purified SynDLP lacks salts like NaCl/KCl. Is the oligomerization affected by salt concentration? How do the oligomers behave under salt concentrations that mimic Cyanobacterial milieu? Perhaps a size exclusion trace and/or a micrograph could help.

3. “Isolated SynDLP forms oligomers of approx. 40–50 molecules in solution in the absence of nucleotides and/or membranes (Fig. 1b, Fig. 2a),...”

The authors suggest that SynDLP^{PRN}-AAAA forms a dimer based on their findings from analytical gel filtration. I agree that it appears that the mutant no longer forms oligomers, but this reviewer is not convinced that it is forming solely a dimer. The analytical gel filtration peak is rather broad and has a shoulder, suggesting a continuous range of oligomeric states. Which fraction of the peak the authors used for cryo-EM sample preparation?

This reviewer was also puzzled that the authors do not see any dimer structures after collecting a cryo-EM dataset of the sample. With the lower limit for cryo-EM structure determination being 52 kDa, 174 kDa protein assemblies should be visible on the micrographs and appear in 2D classification if the sample is relatively homogeneous. It appears to me that while the monomer forms correctly (as indicated by CD and disulfide bond formation), it self-assembles in different states. A 174 kDa dimer should run as a distinct band on a native PAGE gel. Further native PAGE or a SEC-MALS analysis of the sample would be helpful.

Membrane remodeling

4. “...While, in several cases, the formation of membrane tubes has been observed upon the addition of a DLP to lipids, we did not yet succeed in observing the formation of such structures. Yet, we observed a membrane-fusogenic activity of SynDLP, as has been observed previously also for other BDLPs...”

This reviewer feels strongly that further characterization of the membrane interaction, its dependence on SynDLP oligomerization and remodeling behavior will help improve the manuscript. Could the authors include cryoEM micrographs of liposomes incubated with SynDLP? This will be helpful to understand how SynDLP oligomers arranged on the membrane surface and the nature of membrane deformations (if any) they cause. Also, could the authors describe in more detail the specific experimental setup when they were studying membrane tubulation?

It appears that the authors only tested SynDLP interaction with 100 nm LUVs. Since many membrane-shaping proteins are sensitive to membrane curvature and protein to lipid ratios, have the authors tested other LUV diameters as well as protein to lipid ratios? A co-floitation assay or negative stain images of empty LUVs compared to SynDLP in the presence of LUVs could be a nice way to show further relevant insight into protein incorporation or interaction with LUVs.

Finally, the FRET-based fusion assay is not convincing to this reviewer. Since FRET signals will also report on proximity or LUV-LUV apposition, it could give false positive for fusion, when in fact, no lipid mixing has occurred. In addition, the fusion curves in Fig. 7a are steadily increasing instead of plateauing. In Fig. 7b, the initial fusion rate is very low while the error bars are rather large. A more conclusive fusion assay

with a lipid dye or content-transfer would strengthen the conclusions. Is the fusion activity change observed stimulated by nucleotide addition? Does this signal change with the mutants (e.g. HPRN-AAAA, Cys mutant) presented? Experimental evidence of SynDLP incorporation into vesicles (perhaps by co-floitation) would also be helpful.

Minor comments/revisions:

In the introduction, this reviewer would appreciate some general information on the thylakoid architecture in Cyanobacteria (for example, the kind of membrane remodeling events occur - biogenesis, fusion and fission?).

5. "SynDLP is functionally expressed in vivo (Supplementary Fig. 1)."

This sentence may be misleading. The authors describe SynDLP overexpression in *E. coli* in the Methods and Materials section, however, they show SynDLP expression in *Synechocystis* sp. PCC 6803 in Supplementary Figure 1. It would be helpful if the authors could clarify which method was used for all subsequent experiments. Moreover, the term "functionally" is confusing as it suggests that the protein is functional in vivo, however, most functional experiments appear to have been carried out with the purified protein in vitro.

In Supplementary Figure 1, the authors claim to have generated a SynDLP knock-out strain. However, there is a faint band at the same height as the WT band (around 110 kDa) on the western blot. It would be helpful to include confirm the knockout with some form of sequencing data.

Cryo-EM structure resolution reporting: While there understandably is a large resolution gradient across the structure, the clash score in the PDB validation report is still rather high given that roughly 1/3 of the structure is at 3-3.5 Å resolution and permits correct fitting of side chains according to Fig. 2d. In addition, it would be helpful to know if the GFSC resolution was determined with or without masking.

The presentation of the cryo-EM reconstruction could use edits to help a reader better appreciate the work. These include:

6. Including a complete processing workflow, including details such as 2D class number (with example), 2D templates for picking (if used), parameters for ab-initio starting model creation, use of masking, etc.

7. In figure 2 (b), please indicate the side, top and slice views described in the legend also in the actual figure.

8. In figure 2 (e), please label “hinge 1” and “hinge 2” that are discussed in the text in the structure

9. The oligomerization interfaces would benefit from:

a. Preparing a cartoon model of the tetramer to orient the reader, designating the subunits for e.g. subunit A, B, C, D

b. Depicting the side and top views of the cartoon, place the actual structures next to them.

c. In both the cartoons and in the actual atomic models (top/side views), depicting the interfaces 1, 2, 3, GD-GD, GD-stalk and GD-BSE contacts, and specifically indicating if these contacts are between A and B or A and C etc.

Please clearly indicate the location of interfaces (in supplementary figures 4 and 5), so the consequence of the HPRN → AAAA mutation will also be clearer for a potential reader (supplementary figure 6)

10. In the discussion, “...Noteworthy, in contrast to other BDLP structures, such as the structure of NpBDLP that is most closely related to fusion DLPs, the here presented SynDLP oligomer structure resembles a fission DLP structure, similar to classical eukaryotic Dynamin or Drp1, that has not been observed in bacteria before...”

Could the authors elaborate what the resemblance and differences compared to the fission DLP, NpBDLP? A gallery comparing the structures might be helpful.

11. Intramolecular disulfide bridge in the BSE domain:

The intramolecular disulfide bridge and its conservation in the cyanobacterial KGK clade is intriguing and suggests a potential significance. This is an interesting observation, but it would benefit from a more thorough explanation of the author’s observations. While there is a difference in GTPase activity in Figure 3g, the difference seems very low (and the activity still very high) compared to the >100-fold change in GTPase activity upon DLP oligomerization when binding to membranes. Given the reducing environment of the cytosol, it would be helpful to know more about the redox environment of SynDLP to understand if formation of this disulfide bond is reasonable in vivo.

12. In the discussion, “...It remains to be shown how SynDLP oligomerization and futile GTP hydrolysis might be prevented in vivo, likely by accessory proteins...”

The authors assume here that the GTP hydrolysis in vivo is futile, however this reviewer is not convinced there is sufficient evidence to conclusively reach this conclusion. The authors also raise the idea of accessory proteins. It may be worthwhile here to comment on the region/residues in synDLP that senses negatively charged lipids, or what role a potential adaptor protein might play if there are no obvious residues for this type of interaction.

Reviewer #4 (Remarks to the Author):

Summary:

The authors report elegant structural, biochemical, and biophysical experiments that support a role for SynDLP, a dynamin-like protein from cyanobacterium *Synechocystis*, in fusing membranes. SynDLP forms oligomers in solution, in the absence of nucleotides or lipids. A 3.7 Å structure of the SynDLP oligomer was determined by cryo-electron microscopy. Similar to other dynamin like proteins, three interfaces enable oligomerization of the protein. Interestingly, a unique intramolecular disulfide bridge was observed in the bundle signaling element, BSE, which was determined to be important for the GTPase activity of SynDLP. SynDLP was found to bind negatively charged lipids of the thylakoid membrane, specifically, phosphatidylglycerol, and facilitates membrane fusion potentially by membrane destabilization via intercalation. This work presents a dynamin like protein, SynDLP, with unique features and mechanisms of membrane fusion by dynamin like proteins that would be fascinating to explore in future studies.

Minor concerns:

1. Could the authors comment on the essentiality of SynDLP in the lifecycle of *Synechocystis*? Or provide evidence of defective membrane repair in the cyanobacteria with mutant or knocked out SynDLP?
2. Could the authors identify putative lipid binding sites/residues in SynDLP?
3. Membrane fusion dynamins typically have a transmembrane region, with the fusion dynamin proteins bridging opposing membranes. Could the authors discuss or propose a mechanism by which SynDLP accomplishes membrane fusion without a clearly identified transmembrane domain? Or, Could the

authors show electron micrographs of SynDLP organized on liposomes (LUV)? It will be informative to see how the protein is organized. That is are the liposomes tethered, tubulated, or destabilized?

4. The authors suggest SynDLP is unique among dynamin like proteins in not requiring nucleotides or lipids to form oligomers. However, dynamin, the founding member, is also observed to form oligomers in the absence of lipids or nucleotides (PMID: 7877694).

5. SynDLP is a GTPase, and it would be informative to show how its membrane fusion activity is affected as it proceeds through the GTPase cycle of nucleotide binding and hydrolysis.

6. How is the SynDLP oligomer's assembly and disassembly regulated? The current oligomeric structure was under low ionic conditions ~ 7 mM salt. How does nucleotide binding and hydrolysis affect assembly of the oligomer? Could the authors discuss what triggers SynDLP, assembly that presumably occurs during membrane repair.

7. The quality of the structural data is excellent. However, could the authors comment on the presence of rotamer outliers?

Response to reviewer's comments

Reviewer #1:

The manuscript by Gewehr and colleagues describes the structural analysis of a cyanobacterial dynamin-like protein. The authors used cryo-EM to solve the oligomeric structure of SynDLP with a 3.7 Å resolution. SynDLP form short oligomeric filaments of around 50 monomers. The oligomerization is stabilized via stalk interfaces, and thus resembles the oligomerization of eukaryotic dynamins. The authors identified a unique cysteine bridge in the BSE domain of the protein. The cysteines involved seem to be conserved in certain cyanobacterial clades. Removal of the disulfide bridge by mutation of the cysteines reduces GTP hydrolysis activity of SynDLP. Although the overall assembly of the SynDLP oligomer resembles that of classical dynamin, the contacts within the adjacent GTPase domains is different. In line with a different assembly, the basal GTPase activity of SynDLP is comparatively high. As shown for other bacterial DLPs before, SynDLP is able to fuse liposomes *in vitro*, as shown by a FRET-based assay.

This manuscript provides the first oligomeric structure of a full-length bacterial dynamin in solution. Therefore, this work is an important addition to our existing knowledge. The experiments are carefully executed and the data are of high quality. There are only some minor points that should be addressed by the authors in a revision.

- The structure was solved in the absence of nucleotide. Did the authors check for absence of nucleotide after purification?
In absence of externally added GTP, the purified protein shows no GTPase activity, even when a GTP regenerating system is present (Fig. 3g+4g+6, Supplementary Fig. 11c). Thus, we are confident that no significant amount of GTP has been co-purified. We now write on page 12 in lines 229-231: "As both proteins showed no residual (measurable) GTPase activity at 0 mM GTP (negative control without nucleotide; first data point in each curve) although a GTP regenerating system is added, we conclude that no significant amount of GTP/GDP has been co-purified with the proteins."
- Did the authors try to get a cryo structure with added GTP and magnesium?
Of course, we now try to get structures of SynDLP with bound nucleotides. Yet, these structures are beyond the scope of the current work.
- Line 42: Bacterial dynamins were described before the first structure was reported by Low and Löwe. A bioinformatic study identified bacterial DLPs as early as 1999 (DOI: 10.1016/s0962-8924(98)01490-1). Lines 71/72: This sentence is redundant with the one in line 42. Again, the bacterial DLPs have been identified (by bioinformatics) before.
We thank this reviewer for bringing this to our attention. We modified the text accordingly and cite this reference now on page 3 in lines 40-41: "Similarly, Dynamins and Dynamin-like proteins (DLPs) were originally assumed to be eukaryotic inventions until in 1999 a bioinformatic study predicted the existence of bacterial DLPs (BDLPs)⁹."

- Figure S1: What are the other bands – cross reactions from the antibodies? **As suggested, we now state in the figure legend (legend of Supplements Fig. 1, lines 42-43): “The lower bands at ca. 55 kDa resulted from cross-reactions of the antibodies used for immunoprecipitation and Western Blot.”**

- Line 102: “code” might be replaced by “encode”
This is corrected.

- Line 126. A brief explanation why the GD domain have higher flexibility would help. Generally, the GD domain is a well-structured domain, but I assume the authors mean that the GTPase domain may undergo rigid body movements and hence the dynamics? **As suggested, we explained this in more detail. We now write on page 6, lines 120-122: “The local resolution varied from 3.0 to 3.5 Å at the stalk domain to 5.0 to 7.0 Å at the GD, presumably due to tighter contacts between the well-packed stalk domains and looser contacts between adjacent GDs.”**

- Figure 5: Membrane binding measurements with Laurdan GP is not ideal. I agree that binding of a protein to the membrane changes the GP values, but this is a very indirect measurement. Laurdan GP will also vary in membranes with different lipid compositions. A simple assay would be co-sedimentation, but ideal would be a more quantitative interaction method such as SPR. The manuscript would greatly benefit from a better description how SynDLP binds to the membrane. Based on the structure the membrane binding paddle and the residues involved can likely be identified. The authors should at least elaborate more on the membrane binding of SynDLP.
We entirely agree that the exact lipid composition, the buffer etc. can already influence the initial Laurdan GP value. Because of this, we present Δ GP values, i.e., the difference between +/- added protein. As in none of the measurements the maximum GP value was reached, we believe that the measurements allow us to properly analyze the membrane binding propensities.
Yet, in the manuscript we present and discuss another method to monitor membrane interaction (SFG spectroscopy). We used the SFG data to describe some more details of the SynDLP-membrane interaction (page 17-19, lines 330-374, Fig. 5c+d, Supplementary Fig. 15). Importantly, the SFG data clearly show that the protein partly integrates into the membrane and not just attaches to the membrane surface, plus the protein binds in a highly ordered way. Moreover, the SFG data indicate that SynDLP binds the membrane surface in a fast process followed by further structural rearrangement in a slower process. We added this information to the manuscript (page 19, lines 375-385, Supplementary Fig. 16). We also investigated GTP-dependent membrane interaction of SynDLP via SFG spectroscopy, and these measurements indicated no significant differences in the conformation of membrane-bound SynDLP (page 19, lines 386-390, Supplementary Fig. 17), which is in line with the GTPase activity of SynDLP observed to be not stimulated by liposome addition (Fig. 6, Supplementary Fig. 11d).
We would like to mention that we already tried measuring membrane-binding more quantitatively using SPR as well as QCM. Unfortunately, these measurements turned out to be not straightforward, most likely due to the (various) oligomeric structures of SynDLP in the sample. Thus, while we clearly plan to study membrane

interaction of SynDLP more quantitatively in the future, this is beyond the scope of the current manuscript (and was not doable within a revision time of three months). Furthermore, as mentioned by this reviewer, the now solved SynDLP structure enabled us to suggest potential membrane-binding regions. We have identified two putative “paddle” regions at the tip of the stalk domain (Supplementary Fig. 14), based either on the oligomer structure (Supplementary Fig. 14a) or the monomer structure (Supplementary Fig. 14b). We now generated mutants where the putative membrane-binding “paddles” are mutated, purified the proteins (Supplementary Fig. 14c) and compared the membrane-binding propensity of the mutants with the wt protein at identical lipid and buffer compositions, using Laurdan fluorescence spectroscopy (Supplementary Fig. 14d). As the ΔGP values of the mutants showed no difference to the wt, we concluded that the two mutated sites are not (solely) responsible for membrane interaction. We added information about the mutated constructs as well as the results of the Laurdan experiment to the manuscript on page 16-17, lines 324-329 and now write: “In the here solved SynDLPs structure, we recognized two putative MIDs at the tip of the stalk based either on the quaternary (Fig. 1h, Supplementary Fig. 14a) or the tertiary structure (Supplementary Fig. 14b). However, when these regions were mutated, the isolated recombinant proteins still interacted with DOPG containing LUVs similar to SynDLP wt (Supplementary Fig. 14c+d), and thus these regions (alone) are not responsible for membrane binding of SynDLP. Potentially, either other regions of the protein or a larger area involving multiple SynDLP parts are responsible for membrane interaction.” The experimental results are shown in the supplement.

- Line 312: 30 min (instead of 30 in)
We apologize for the typo, which is now corrected.
- Figure 7: Did the authors add nucleotide or magnesium to the fusion assays? **As stated in the manuscript, 5 mM Mg^{2+} was present in the fusion assay, but no nucleotides. Yet, we now show in the new Supplementary Fig. 20 that fusion is essentially not altered by the presence vs. absence of GTP.**
- Line 388: Please introduce the term KGK clade. The paper will be read not only by cyanobacterial specialists. Is there something special in the KGK clade that can be related to the disulfide bridge being necessary?
We agree that this was not self-explanatory and further commented on this in the manuscript. On page 22, lines 452-454 we now write: “Cyanobacteria contain several putative genes encoding DLPs, which can be grouped into different clades depending on their genetic context. E.g., in the KGK clade of cyanobacterial DLPs typically a protein containing a KGK domain is encoded downstream of the DLP⁴⁴.”
- Discussion: Generally, fission and fusion of membrane is mechanistically related and involves a hemifusion state. Therefore, lipid mixing (this is what is shown in Figure 7) is not necessarily a proof of fusion in an in vivo setting. The eukaryotic Dyn2 is also a fission GTPase in vivo, but in vitro it can induce lipid tethering (and mixing) under certain experimental conditions. The discussion should therefore be slightly tuned down. **We entirely agree with this reviewer and now also clearly state on page 21, lines 427-429 and page 25, lines 538-539 that we might see membrane destabilization, which**

results in liposome fusion in the chosen in vitro assay. As suggested, we toned down the statements and discussion about the fusogenic activity of the protein.

Reviewer #2:

Gewehr et al report the cryoEM structure of oligomerized SynDLP, a dynamin-related protein from the cyanobacterium *Synechocystis* sp PCC 6803. Based on their structural analysis, they performed a biochemical study to conclude that SynDLP uses an uncommon activation mechanism for GTP hydrolysis.

The structural work is noteworthy and remarkable since SynDLP appears to be the first bacterial dynamin-like protein that is closely related to eukaryotic dynamin and Drp1, two key molecules of membrane remodeling in higher eukaryotes. However, the structural analyses and representations are not sufficient to understand the architecture and significance of the new model. Also, the biochemical data set appears incomplete; additional experiments are required to support the conclusions. Some data may not be correctly interpreted.

Major:

- 1.) This is the first structural data of this type of bacterial dynamin-like protein, but from the current presentation, it is difficult to understand the DynDLP architecture. Structural figures are not properly labelled and are too small to recognize any details. The authors have to put more efforts to explain the newly determined structure to the reader. In particular, Fig. 2 should contain a structure-based domain representation of DynDLP. A topology plots would be helpful to understand the detailed architecture. Fig. 2e should be enlarged and properly labelled (GTPase domain, stalk, BSE, the two hinges including the proline residues, the membrane-binding site, nucleotide binding site, some secondary structure elements, etc). The four-helix architecture of the BSE should be properly displayed. Higher magnifications of functionally relevant sites should be provided, in which key amino acids are properly labelled. Although the resolution of the cryoEM density is relatively low in some parts, the fitting with the AF2 model, as done in this study, should allow the identification of key amino acids in some of the interfaces. Accordingly, molecular details of all assembly interfaces should be provided and again properly labelled (unlike in Supp. Fig. 6). The disulfide bridge needs to be shown in detail and labelled in a magnified view (Fig. 3a is not helpful in this regard).

We improved the clarity of the structural data (EM) presentation, as suggested by this reviewer in an updated main Fig. 1: We added a topology plot to show the detailed domain architecture (Fig. 1f) and we enlarged the monomer structure and properly labeled the domains, the two hinges and secondary structure elements. Furthermore, we now show higher magnifications of the functionally relevant HPRN loop, putative lipid-binding sites investigated in this study and the nucleotide-binding site with labeled key amino acids (Fig. 1h). The four-helix bundle of the BSE domain + the disulfide bridge between BSE1 and BSE3 are now shown more clearly in the magnified section of Fig. 3a. Molecular details of the three oligomerization interfaces are now displayed in Supplementary Fig. 7.

- 2.) From what I understand, SynDLP could be the long sought-after bacterial ancestor of eukaryotic dynamin, MxA and Drp1; the other dynamin-related proteins reported so far, show somewhat different assemblies and modes of action. However, to make this important claim, a thorough structural comparison of SynDLP to other reported dynamin proteins from bacteria and eukaryotes has to be performed. rmsds of isolated domains (in particular the stalk and BSE) should be reported, topologies and assembly interfaces should be compared, surface conservation plots should be provided in a separate main figure. At the moment, the available structural data are not fully mined. **As suggested, we inserted an additional section (page 8-9, lines 158-176) including a new main figure (Fig. 2) showing the structural comparison in addition to reviewer #3' point 10 request on NpBDLP and based on RMSD values obtained from structural alignments. We now show a surface conservation plot of the SynDLP structure (Fig. 2a) and compare isolated SynDLP domains to isolated domains of the bacterial DLPs NpBDLP and MslniA as well as the eukaryotic representatives Dyn3, MxA and Drp1. Based on alignments of isolated stalk domains (Fig. 2c) as well as BSE domains (Supplementary Fig. 10) suggested a closer structural similarity of SynDLP with eukaryotic DLPs. This was supported by the similar mode of assembly identified in SynDLP and some eukaryotic DLPs. We now show this comparison in a new figure (Supplementary Fig. 8a). We concluded from this analysis that SynDLP might indeed be a bacterial ancestor of the eukaryotic DLPs dynamin, MxA and Drp1.**

- 3) Figure 5: The membrane-binding site in dynamin-related proteins is normally localized in a loop or domain at the tip of the stalk. Based on the performed analysis, it seems reasonable to assume that it involves a patch of positively charged amino acids in the related SynDLP loop at the tip of the stalk. It will be straightforward to identify and describe the membrane-binding site with a small set of mutations, which would strengthen the impact of the study. **As indicated, inspection of the structure allowed us to identify two “paddle” regions putatively involved in membrane binding (see Fig. 1h, comment to reviewer #1). The membrane binding propensity of mutant proteins, where the putative membrane binding sites are mutated, were now analyzed (as mentioned in the response to reviewer #1). As we stated in the response to reviewer #1, the mutant proteins showed no significant difference in membrane interaction compared to the wt. Thus, unfortunately identification of the SynDLP membrane binding site(s) is not as straightforward as initially assumed, and the regions to interact with membranes not turn out to mediate membrane interactions (alone). Nevertheless, we now presented the new data in the manuscript (new Supplementary Fig. 14) gained using SynDLPs with the mutated regions.**

- 4.) The GTPase domain is the most highly conserved region in dynamin proteins. In fact, essentially all examined dynamin-related and septin-related proteins use a GTPase domain-mediated dimerization mechanism via the ‘G-interface’ for stimulated GTPase activity. Furthermore, the G-interface is structurally well characterized in various dynamins, but in many cases, it is observed only in the presence of GTP/GDP-AIF3, not without nucleotide. Given the close structural resemblance of SynDLP and dynamin, it is reasonable to assume that SynDLP also uses a related dimerization-dependent GTPase mechanism. On the other hand, it has already been described that the assembly of

dynamamin superfamily proteins via the stalk or GED can indirectly affect GTP hydrolysis rates (see for example, Gao et al, Nature 2010, for related studies on dynamamin-like MxA). The ‘non-canonical’ GTPase domain-BSE contacts shown in Supp. Fig. 4 may have a similar indirect effect on GTPase activation. Thus, the notions that ‘non-canonical’ contacts in the GTPase domain stimulate GTP hydrolysis (title, abstract, line 216) and the idea of a GTPase-regulating function for the BSE (line 95) are likely misleading. Some further experimental analysis is warranted: How is the GTPase activity affected for mutants in the ‘non-canonical’ GTPase domain interfaces (rather than relying on mutations far away from the GTPase domain)?

We replaced the term ‘non-canonical’, as it is rather confusing, and now refer to “transversal” GD-GD contacts (like observed in GD-dimerization in a, e.g., Dynamamin oligomer on a membrane template leading to dramatic increase of Dynamamin GTPase activity) and “longitudinal” interactions (like observed in the SynDLP filament structure between adjacent GDs and especially the intermolecular GD-BSE contacts in the absence of membranes). We completely agree with this reviewer and also recognized that such mutations would help to better understand the activation mechanism. Unfortunately, the resolution in the GTPase-domain was not high enough to identify single residues involved in this interaction. Yet, we were able to recognize the residues involved in the GD-BSE interface on the BSE and now mutated these to alanine (Fig. R1a). Unfortunately, the mutant protein was prone to aggregation (Fig. R1b) and could not be properly analyzed.

Figure R1: Mutation of BSE1 residues contacting the GD. a) Section of the SynDLP oligomer structure. A mutant protein (SynDLP_{BSE1Ala}) was expressed, where the GD-interfacing residues R12, N16, E20, R23, P26, S30, D33, S35, E38, G42 and L45 of the BSE1 domain were substituted by Ala. GD shown in red, BSE in purple, mutated residues in cyan and as ball and sticks (in monomer 1). Monomer 1 is shown as ribbon structure, monomer 2 as surface representation. b) Analytical gel filtrations of SynDLP wt (black) and SynDLP_{BSE1Ala} (red) demonstrated the formation of aggregates in case of the mutant protein.

We briefly mentioned this now in the manuscript on page 24-25, lines 514-516 and write: “While we created a mutant where eleven residues in the BSE1 (R12, N16, E20, R23, P26, S30, D33, S35, E38, G42, L45) that contact the GD were replaced, the resulting protein was prone to aggregation and not suitable for subsequent analyses.”

- Does SynDLP display a protein concentration-dependent GTPase stimulation? **New GTPase activity measurements with varying protein concentrations were performed and we added the results to the supplements (Supplementary Fig. 11a).**

We observed a concentration-dependent GTP turnover rate but only at SynDLP concentrations <0.3 μ M and, thus, we now write on page 12, lines 214-216: “Typically, the GTPase assay was performed with 0.5 μ M protein, as SynDLP shows a concentration-dependent GTPase activity at low protein concentrations, yet reaching a plateau at protein concentrations >0.3 μ M (Supplementary Fig. 11a)”.

- **Can GTPase activity be abolished by a point mutation in the G-interface? This would unambiguously demonstrate related GTPase mechanisms of dynamin and SynDLP. As suggested, we now mutated a conserved residue in the active site (K61A) known to be crucial for the GTPase activity of other DLP family members. As expected, the GTPase activity of the variant was essentially abolished. The determined GTPase activity is compared to the wt protein and now shown in the manuscript in the Supplementary Fig. 11c. Furthermore, we now write in the text on page 12, lines 219-221: “A mutation of a conserved residue in the P-loop (K61A) considerably reduced the GTPase activity, as previously observed for Dynamin and other DLPs, which demonstrated a related GTPase mechanism (Supplementary Fig. 11c)”**

- **5.) GTPase activity: Dynamin family proteins use a dimerization-mediated GTPase mechanism which is based on different assumptions from the classical Michaelis-Menten kinetics. In particular, a lower observed Km value is not necessarily related to an altered nucleotide binding affinity, but it could also be explained by a different assembly kinetics of the GTPase domains in oligomerized vs. dimeric constructs. How should a mutation in the stalk directly affect the nucleotide-binding affinity in the GTPase domain? A direct comparison of nucleotide-binding affinities, for example using mant-nucleotides, should be performed to make such a claim. As suggested, we now determined nucleotide-binding affinities of wt SynDLP as well as of the oligomerization impaired variant (SynDLP_{HPRN-AAAA}) and the other characterized mutant (SynDLP_{C777A}) via fluorescence spectroscopy using MANT-GTP. When determined with this method, the nucleotide binding affinities of the analyzed proteins did not alter significantly. The results were added to the manuscript (Supplementary Fig. 12) and we now write on page 12, lines 226-229: “GTP binding affinities of SynDLP wt and C777A were determined via fluorescence anisotropy measurements using the GTP analog Mant-GTP (Supplementary Fig. 12). Based on this assay, both proteins have similar nucleotide-binding affinities in the three-digit nanomolar range” and concerning the HPRN-AAAA mutant on page 15, line 286-288: “The GTP binding affinity of the mutant appeared not to be significantly affected as indicated by a Mant-GTP binding assay (Supplementary Fig. 12).”**

- **6.) As already stated in the manuscript, the reported ‘basal’ GTPase activity of SynDLP appears unusually high when compared to other dynamin-related proteins. Were only initial rates considered for the Michaelis-Menten kinetics in the 2-3h measurement period reported (which seems very long for such a fast reaction)? We apologize for obviously not having explained the evaluation of our data in enough detail in the manuscript. Only the areas with highest slope in the A_{340 nm} decrease were considered for the calculation of the GTPase activities. E.g., for the highest tested GTP concentration (5 mM) the signal decrease stopped after 15-20 min when all NADH was oxidized, and thus the activity was determined based on the first few minutes. Yet, we needed to measure for up to 3 h just to make sure that**

the highest slope in signal decrease is trapped for all measured GTP concentrations. We now explain this a bit more in detail in the manuscript on page 29 in lines 623-625: “The absorption at 340 nm was observed with a microplate reader (FLUOstar® Omega, BMG Labtech GmbH, Ortenberg, Germany) at 30°C over 2-3 h to make sure that even at the lower GTP concentrations all NADH was oxidized.”

- **The inclusion of 5 mM MgCl₂ could prevent any liposome-stimulated GTPase activity due to clustering of the lipid head groups. GTPase measurements should be repeated with liposomes in the presence of 0.5 mM MgCl₂ and 150 mM KCl instead of NaCl since dynamin proteins use a K⁺-dependent GTPase mechanisms (Dyda et al, Nature 2010). As suggested, the GTPase activity assay was now additionally measured in the presence of 0.5 mM MgCl₂ and in the presence or absence of DOPG LUVs to estimate any influence of putative lipid head group clustering. This new data is now shown in Supplementary Fig. 11d. Yet, we clearly need Mg²⁺ in the measurements due to the GTP. The results were included in the manuscript. We refrained from measurements in presence of 150 mM KCl and at the same time the absence of NaCl, as SynDLP displays a high GTPase activity already in presence of NaCl and these conditions would not be comparable to all other presented experiments.**

- **7.) Fig. 7: The structural similarity of SynDLP and dynamin/Drp1 points to a related mechanism of the proteins in membrane constriction. From this point of view, the membrane fusion properties of SynDLP reported in Fig. 7 appear highly surprising. However, the assay on its own seems not sufficient to support the conclusions of SynDLP being a fusogen. The undefined oligomeric assemblies of SynDLP observed in solution may just induce non-specific liposome aggregation, which may also explain the fluorescent increase in the applied assay. The authors should complement their experiments with negative-stain EM analyses of SynDLP and liposomes in the absence and presence of GTP (again using maximally 0.5 mM MgCl₂) to observe putative fusion or fission states. Alternatively, a light microscopy-based liposome tethering assay could be employed, similar as shown for IniA in Wong et al, Nat Comm 2019 – in fact, this article also contains nicely labelled structural figures for comparison. In fact, the here used FRET-based assay would not show altered FRET when liposomes merely cluster, as the dyes are not diluted by liposome clustering and the mean distance between the dyes is not altered. Thus, the used FRET-based assay in the end can solely be interpreted by liposome fusion, which we now support by new measurements showing increasing sizes of the sample (Fig. 7a). Yet, we entirely agree with this reviewer that the determined fusogenic activity is surprising and toned down the statements about SynDLPs fusogenic activity. As also mentioned in response to reviewer #1, we now state that we likely observed membrane destabilization, which results in membrane fusion under the chosen in vitro conditions. Yet, membrane destabilization in the end is required for both, membrane fusion as well as membrane fission.**

- **8.) A detailed description of the cellular function of SynDLP may be beyond the scope of the manuscript. However, since the SynDLP ko strain has already been generated (Supp. Fig. 1), it would be neat to at least describe whether the thylakoid membranes appear altered in this mutant or whether photosynthesis is possibly affected? To suggest putative**

roles of SynDLP in phage infection or membrane repair (line 441-447) seems too speculative without any supporting experimental data.

Thus far, we did not analyze the mutant strain in detail, as we concentrated on the now presented structural analyses. Nevertheless, a detailed analyses of the mutant's phenotype clearly is "on the list", yet beyond the scope of the present manuscript. However, first results indicated no severe phenotype of the knock-out mutant, at least under standard growth conditions. We added some information to the legend to Supplementary Fig. 1, where we now write in lines 33-35: "The deletion strain grows like the wt and did not show severe defects in photosynthetic performance. Thus, our initial in vivo experiments indicated no altered phenotype of the SynDLP knock-out strain, at least under standard growth conditions."

Minor:

- Description of the results could be shortened at several positions. For example, line 105-109 could go to the Methods, Fig. 1a could go to the supplement (ideally with some further information on the protein purification), Fig. 1b could be combined with Fig. 2. The reduced melting temperature of the C777A mutant is maybe not a huge surprise so this paragraph can be shortened (line 149-167). Also the ANS assay and the fitting procedures of the curve do not to be explained in depth in the results (line 169-179) and the description of the biochemical characterization of the interface-3 mutant (line 229-249) could be shortened. This would leave more space to elaborate on the structural analysis.

As suggested, we shortened the text at several places. We also split former Fig. 1 and now show an SDS-PAGE image of purified SynDLP in Supplementary Fig. 2 and combined former Fig. 1b with former Fig. 2. Information about the protein purification were already given in the Material and Methods section.

- Fig. 2a: What is the radius that the SynDLP oligomer would embrace? It seems rather a low curvature radius compared to dynamin?

Indeed, the curvature is very low compared to other oligomeric assemblies of dynamins. We briefly commented on this in the manuscript and now write on page 6, lines 112-115: "When we visualize SynDLP by cryo-EM using prepared plunge-frozen vitrified specimen, we observed short oligomeric filaments of bent half-moon shape (Fig. 1b) with typical lengths of about 100 nm and a curvature radius of approximately 50 nm." And on page 6, lines 127-128: "Interestingly, the curvature of the SynDLP oligomers is rather low compared to similar assemblies of other DLPs (reviewed here²⁰)."

- CryoEM analysis:

Is there any chance to improve the resolution of the GTPase domain by local refinement and masking (as for example, in Chaaban and Carter, Nature 2022)? In fact, we already attempted to improve the resolution of the GTPase-domain by local (focused) refinement (follow in the new Supplementary Fig. 5a of the processing workflow). Although the additional refinement did not improve the nominal resolution of the GD, it increased the interpretability of this area. In this way, we were able to identify individual helices and beginning separation of beta

strands (see new Supplementary Fig. 5b), which finally allowed us to flexibly fit the AF model of the GD into the density.

- Fig. 5: SFG spectroscopy appears to be a rather specialized method for monitoring protein-membrane interactions. For example, what do we learn from Fig. 5d? this analysis provide useful information to the current manuscript?

This measurement quantitatively demonstrates that the protein binds to the membrane in a highly oriented way. Furthermore, the surface pressure measurements show that the protein partly integrates into one monolayer and not just binds to the membrane surface. In the revised manuscript we elaborated more on these observations on page 18-19, lines 365-390 (for more details see also response to reviewer #1).

- Typos

Line 140: Supp. Fig. 2b does not exist.

We apologize for the typo, which we corrected in the revised manuscript.

Reviewer #3:

Major comments:

Nucleotide State

- 1. "...Yet, in contrast to other BDLPs, SynDLP forms high-molecular mass oligomers already in complete absence of lipids and/or nucleotides..."
It is unclear how the authors conclude that lipids and/or nucleotides are "completely" absent, since nucleotides or native lipids might have co-purified with the protein. With a local resolution distribution between 3.7 Å to 7 Å, it would be challenging to detect lipid density in certain areas of the structure. Do the authors have other evidence to support this statement? Have the authors checked for lipids or nucleotides by mass spectrometry or other means?

We apologize for the misleading wording. Of course, we cannot completely exclude that any lipids were co-purified with the proteins. Yet, similar studies have shown that interaction of dynamins with membrane surfaces results in protein oligomerization and activates the protein. Thus, the statement actually refers to SynDLP interaction with membranes (not lipids). As described already in the answer to reviewer #1, the purified protein does not have any residual (measurable) GTPase activity when a GTP regenerating system is added, and thus, no (significant) amounts of GTP or GDP were copurified with the protein. To address the points raised by this reviewer concerning the potentially co-purified lipids and nucleotides, we now state this more clearly in the manuscript on page 6, lines 108-110 and write: "Yet, in contrast to other BDLPs, SynDLP forms high-molecular mass oligomers already in complete absence of an externally added membrane template and/or nucleotides."

- 2. In the discussion, "...It remains to be shown how SynDLP oligomerization and futile GTP hydrolysis might be prevented in vivo, likely by accessory proteins..." The authors assume here that the GTP hydrolysis in vivo is futile. This reviewer has concerns about the GTPase assay setup. A negative control of protein without nucleotide is lacking in the final figures.
Every measurement series has one data point measured at 0 mM GTP that is subtracted by the blank without protein (= negative control without nucleotide). We apologize for obviously not having explained this in enough detail and explained this more clearly in the manuscript now on page 12, lines 229-231: "As both proteins showed no residual (measurable) GTPase activity at 0 mM GTP (negative control without nucleotide; first data point in each curve) although a GTP regenerating system is added, [...]."

- In addition, unlike all other experiments, the authors conduct the GTPase activity assay in the presence of salt (150 mM NaCl, 7.5 mM NaCl).
Actually, the ANS-FTSA, fusion, SEC and the SFG experiments were all performed at the same salt conditions as the GTPase activity assay. Some assays (CD spectroscopy, Laurdan fluorescence spectroscopy) were performed without salts due to disturbing effects of the salts on the measurements. Cryo-EM analysis of SynDLP oligomers was performed without NaCl, as the SynDLP filaments were more ordered and elongated under these conditions, which was beneficial for the subsequent data procession. We therefore now write on page 6 in lines 115-117: "Noteworthy, in contrast to biochemical assays, EM micrographs were acquired in the absence of NaCl, as the oligomeric filaments appeared longer and more defined under these conditions (Supplementary Fig. 4) and therefore more suitable for structural analysis."

- Given the sensitivity of DLP self-assembly to changes in salt conditions, it would be interesting to know if the authors observed any differences in GTPase activity or SynDLP oligomerization under the same salt conditions used for the other experiments (for cryo-EM: 5 mM MgCl₂, 7.5 mM KCl; for Laurdan fluorescence, CD and purification: no salt).
As Mg²⁺ and K⁺ are prerequisites for a (dynamin-like) GTPase activity, the GTPase assay can usually not be performed without these two salts. However, we determined the GTPase activity in absence of NaCl to mimic the conditions used for cryo-EM (Supplementary Fig. 11b). Here, we observed a decrease in the GTPase activity in the absence of NaCl showing the importance of the salt in the active site. We refrained from using SEC to check the influence of low salt conditions on the oligomerization, as these conditions are reported to potentially lead to unwanted electrostatic interactions of proteins and SEC column and in the end jam the column. Nevertheless, the impact of salt on SynDLP oligomerization has been visualized by negative stain EM and the micrographs are now added to the manuscript in the new Supplementary Fig. 4, which shows elongated and more ordered SynDLP filaments in the absence of NaCl than with increasing NaCl concentrations. In fact, this is the reason why we performed cryo-EM micrographs for structure determination in the absence of NaCl.

Oligomerization State

- Related to the nucleotide comments above, the final buffer for purified SynDLP lacks salts like NaCl/KCl. Is the oligomerization affected by salt concentration? **The final buffer to store SynDLP lacks indeed salts since the solubility of SynDLP is reduced at high salt concentrations, e.g., in the presence of 150 mM NaCl. Therefore, the salt-free buffer was needed to prevent protein precipitation after longer storage.**
- How do the oligomers behave under salt concentrations that mimic Cyanobacterial milieu? Perhaps a size exclusion trace and/or a micrograph could help. **Exact salt concentrations in the cyanobacterial milieu are difficult to obtain. E.g., for NaCl, there are publications that at least suggest values ranging around 50 mM NaCl. In the new Supplementary Fig. 4, negative stain EM micrographs in the presence of 50 mM NaCl are added and still show filamentous SynDLP oligomers.**
- 3. “Isolated SynDLP forms oligomers of approx. 40–50 molecules in solution in the absence of nucleotides and/or membranes (Fig. 1b, Fig. 2a),...” The authors suggest that SynDLP_{HPRN-AAAA} forms a dimer based on their findings from analytical gel filtration. I agree that it appears that the mutant no longer forms oligomers, but this reviewer is not convinced that it is forming solely a dimer. The analytical gel filtration peak is rather broad and has a shoulder, suggesting a continuous range of oligomeric states. Which fraction of the peak the authors used for cryo-EM sample preparation? **The whole protein sample (before SEC) was used for cryo-EM sample preparation and not fractions of an SEC separated protein. We now stated this more clearly in the legend of Supplementary Fig. 13, and on page 15, lines 142-143 we now write: “The entire protein sample before analytical gel filtration (see Fig. 4c) was analyzed (no single fractions).”**
- This reviewer was also puzzled that the authors do not see any dimer structures after collecting a cryo-EM dataset of the sample. With the lower limit for cryo-EM structure determination being 52 kDa, 174 kDa protein assemblies should be visible on the micrographs and appear in 2D classification if the sample is relatively homogeneous. It appears to me that while the monomer forms correctly (as indicated by CD and disulfide bond formation), it self-assembles in different states. A 174 kDa dimer should run as a distinct band on a native PAGE gel. Further native PAGE or a SEC-MALS analysis of the sample would be helpful. **The 2D classes of the cryo-EM dataset of the SynDLP_{HPRN-AAA} mutant were already included in the corresponding Supplementary Fig. 13c. Unfortunately, further processing of the data was not possible as the particles could not be aligned in 3D.**

Membrane remodeling

- 4. “...While, in several cases, the formation of membrane tubes has been observed upon the addition of a DLP to lipids, we did not yet succeed in observing the formation of such structures. Yet, we observed a membrane-fusogenic activity of SynDLP, as has been observed previously also for other BDLPs...”

This reviewer feels strongly that further characterization of the membrane interaction, its dependence on SynDLP oligomerization and remodeling behavior will help improve the manuscript. Could the authors include cryoEM micrographs of liposomes incubated with SynDLP?

We now further support the observations from the FRET-based assay by new measurements showing increasing sample sizes (Fig. 7a). And we entirely agree that a more detailed description of the SynDLP-membrane interaction would be desirable. Yet, our initial attempts to quantify the membrane interaction via SPR or QCM (unfortunately) failed, and a substantial amount of work is needed to tackle this question.

- This will be helpful to understand how SynDLP oligomers arranged on the membrane surface and the nature of membrane deformations (if any) they cause. Also, could the authors describe in more detail the specific experimental setup when they were studying membrane tubulation?

It appears that the authors only tested SynDLP interaction with 100 nm LUVs. Since many membrane-shaping proteins are sensitive to membrane curvature and protein to lipid ratios, have the authors tested other LUV diameters as well as protein to lipid ratios? **Tested experimental conditions to induce membrane tubulation were: addition of different nucleotides (GTP, GDP, GMPPnP), different MgCl₂ concentrations, different protein:lipid ratios (2:1, 7:5, 1:1, 1:2, 1:5, 1:10), different lipid compositions (100% DOPG, 70%DOPC/30% DOPG), usage of only fresh protein, purification of protein in the salt-containing buffer. Different LUV diameters were not systematically tested thus far, as liposomes with 100 nm diameter are already curvature stressed and should be prone to tubulation. Nevertheless, we feel that this information is not required and goes beyond the scope of the current manuscript.**

- A co-flotation assay or negative stain images of empty LUVs compared to SynDLP in the presence of LUVs could be a nice way to show further relevant insight into protein incorporation or interaction with LUVs.

We attempted a co-flotation assay of SynDLP together with DOPG LUVs, yet resulted in a diffuse pattern of the protein distribution (not shown). Yet, we now elaborate more on the SFG studies, showing that SynDLP presumably incorporates into the membrane and we now write on page 19, lines 371-374: “The spectra (Supplementary Fig. 15) showed a signal increase for the acyl CD₃ modes, which strongly suggested the acyl chains became more ordered with SynDLP binding, which could be the result of the ordering effect of charge-charge interactions between protein and lipid interface or intercalation of protein side chains into the lipid layer.”

- Finally, the FRET-based fusion assay is not convincing to this reviewer. Since FRET signals will also report on proximity or LUV-LUV apposition, it could give false positive for fusion, when in fact, no lipid mixing has occurred. In addition, the fusion curves in Fig. 7a are steadily increasing instead of plateauing. In Fig. 7b, the initial fusion rate is very low while the error bars are rather large.

A more conclusive fusion assay with a lipid dye or content-transfer would strengthen the conclusions.

Actually, the fusion assay depends on dilution of the FRET pair in an increased available membrane surface area due to fusion of labeled with unlabeled liposomes (Fig. R2).

Figure R2: Membrane fusion assay. Liposomes containing donor and acceptor-labeled lipids were mixed with unlabeled liposomes in a 1:10 ratio. Liposome fusion results in dilution of the dyes, resulting in an increased mean distance and reduced FRET.

Thus, the fusion assay was already performed with a lipid containing a covalently attached fluorophore dye (as now suggested). We repeated the experimental setup of the membrane fusion assay, now leading to higher initial fusion rates whilst the error bars are reduced. We present the new data set in Fig. 7b+c and Supplementary Fig. 18. Nevertheless, to show membrane fusion by other means, we performed light scattering analyzes of liposomes +/-SynDLP to monitor changes in liposome size (Fig. 7a). Increasing sizes in the mixture of liposomes + SynDLP indicated fusion processes. However, we likely observed membrane destabilization leading to the observed membrane fusion under the selected in vitro conditions. Thus, we now toned down statements concerning the fusogenic activity of SynDLP.

- *Is the fusion activity change observed stimulated by nucleotide addition?*
We now performed the fusion assay in the presence of GTP and did not see any changes in the activity. We added this information to the manuscript on page 21, lines 426-427 and show the data in the Supplementary Fig. 20.
- *Does this signal change with the mutants (e.g. HPRN-AAAA, Cys mutant) presented?*
The fusion assay was now performed also with the two mutants HPRN-AAAA and C777A. We did observe only slight differences in case of the HPRN-AAAA mutant. We added the results to the manuscript on page 22, lines 423-425 and write: “The two investigated mutants SynDLP_{C777A} and SynDLP_{HPRN-AAAA} also showed an increased donor fluorescence, yet in case of the assembly-defect SynDLP_{HPRN-AAAA} mutant with a lower efficiency” and in the new Supplementary Fig. 19.
- *Experimental evidence of SynDLP incorporation into vesicles (perhaps by co-flotation) would also be helpful.*
In the Laurdan fluorescence measurements, we determined changes in the membrane structure caused by surface adhesion of SynDLP. Furthermore, SynDLP incorporation into a membrane leaflet was shown in the surface pressure measurements (Fig. 5c). As we did obviously not explain this in enough detail, we added a new Supplementary Fig. 15 and further elaborated on this in the revised manuscript on page 18-19, lines 365-3874 and write: “Interestingly, the resonance near 1738 cm⁻¹ related to the PG carbonyl group remained largely unchanged after SynDLP binding, indicating the lipid head groups remained ordered when interacting with the protein. This supported the assumption of charge interactions

involved in the binding mechanism, since such interactions will align the lipid headgroups in the process. To investigate the interactions of SynDLP with the lipid acyl chains, we recorded SFG spectra in the C–D stretching range using lipids with perdeuterated acyl chains. The deuteration allows monitoring the state of the lipid layer without interference by protein C–H modes. The spectra (Supplementary Fig. 15) showed a signal increase for the acyl CD₃ modes, which strongly suggested the acyl chains became more ordered with SynDLP binding, which could be the result of the ordering effect of charge-charge interactions between protein and lipid interface or intercalation of protein side chains into the lipid layer.”

Minor comments/revisions:

- In the introduction, this reviewer would appreciate some general information on the thylakoid architecture in Cyanobacteria (for example, the kind of membrane remodeling events occur - biogenesis, fusion and fission?).

As suggested, we elaborated on this a bit more in the introduction, page 5, lines 85-88 and now write: “Due to the photosynthetic light reaction, TMs are highly vulnerable to light stress and are continuously remodeled⁴⁹, and thus, proteins mediating membrane remodeling and/or repair via membrane fusion and fission are required.”

- 5. “SynDLP is functionally expressed in vivo (Supplementary Fig. 1).” This sentence may be misleading. The authors describe SynDLP overexpression in *E. coli* in the Methods and Materials section, however, they show SynDLP expression in *Synechocystis* sp. PCC 6803 in Supplementary Figure 1. It would be helpful if the authors could clarify which method was used for all subsequent experiments. **In principle, all experiments were performed using heterologously expressed and purified SynDLP. We stated this now more clearly in the manuscript in the M&M section on page 28, lines 604-606 and now write: “All subsequent experiments were performed with heterologously expressed and purified protein (except the immunoprecipitation experiment in Supplementary Fig. 1, which showed expression of native SynDLP).”**

Moreover, the term “functionally” is confusing as it suggests that the protein is functional in vivo, however, most functional experiments appear to have been carried out with the purified protein in vitro. In Supplementary Figure 1, the authors claim to have generated a SynDLP knock-out strain. However, there is a faint band at the same height as the WT band (around 110 kDa) on the western blot. It would be helpful to include confirm the knockout with some form of sequencing data.

As suggested, we analyzed the deletion strain via PCR and now show this in Supplementary Fig. 1a. Our analyzes confirms that the gene is completely deleted in the KO strain.

- Cryo-EM structure resolution reporting: While there understandably is a large resolution gradient across the structure, the clash score in the PDB validation report is still rather high given that roughly 1/3 of the structure is at 3-3.5 Å resolution and permits correct fitting of side chains according to Fig. 2d. In addition, it would be helpful to know if the GFSC resolution was determined with or without masking.

As stated in the Methods section (line 795), the GFSC was determined with auto-masking by CryoSparc at 3.7 Å. We now added this information to the figure legend of Fig. 1d and the Cryo-EM workflow scheme in Supplementary Fig. 5a, to make it more clear to the reader.

Based on the reviewers request, we describe the clashes in more detail. The clashes are mostly localized in the oligomerization interface around Hinge 1 and result from the oligomer contacts rather than the monomer structure. In this interface, many bulky residues are condensed in close contact. Although the local resolution in that area is 3.5 to 4.0 Å, side-chain densities are not always defined. During the refinement, we balanced correct geometry and clashes and favored geometry restraints over distance restraints as the resolution of our reconstruction is not good enough to justify geometry outliers. Our refined structure has a clashscore of 8.56 while many EM structures at this resolution exhibit higher clash scores. According to MolProbity and the PDB validation reports, clashscore of 8 is within the acceptable range compared to the percentile relative to all EM structures.

- *The presentation of the cryo-EM reconstruction could use edits to help a reader better appreciate the work. These include:*

6. Including a complete processing workflow, including details such as 2D class number (with example), 2D templates for picking (if used), parameters for ab-initio starting model creation, use of masking, etc.

7. In figure 2 (b), please indicate the side, top and slice views described in the legend also in the actual figure.

8. In figure 2 (e), please label “hinge 1” and “hinge 2” that are discussed in the text in the structure

9. The oligomerization interfaces would benefit from:

a. Preparing a cartoon model of the tetramer to orient the reader, designating the subunits for e.g. subunit A, B, C, D

b. Depicting the side and top views of the cartoon, place the actual structures next to them.

c. In both the cartoons and in the actual atomic models (top/side views), depicting the interfaces 1, 2, 3, GD-GD, GD-stalk and GD-BSE contacts, and specifically indicating if these contacts are between A and B or A and C etc.

Please clearly indicate the location of interfaces (in supplementary figures 4 and 5), so the consequence of the HPRN -> AAAA mutation will also be clearer for a potential reader (supplementary figure 6).

We thank this reviewer for these very helpful suggestions and agree that this further improves the presentation. We now improved the presentation of the cryo-EM workflow in a new Supplement Fig. 5a. Concerning the structure representation, we labeled side, top and slice view in the actual figure (Fig. 1c), labeled the two hinges in the monomer structure (Fig. 1h) and prepared a cartoon model which clarifies the arrangement of the monomers within the oligomer by designating the subunits with A and B (Fig. 1g). We also updated a figure to better illustrate the three oligomerization interfaces now with molecular details (Supplementary Fig. 7) and we updated the figure showing the intermolecular GD-GD, GD-stalk and GD-BSE interfaces, where we now indicate, if these contacts are between monomer A, B or C (Supplementary Fig. 22). Additionally, we added a magnification of the HPRN-loop in the main Fig. 1h and hope that this together with Supplementary Fig. 7+13 now clearly illustrate the consequence of the HPRN-AAAA mutation.

- 10. In the discussion, "...Noteworthy, in contrast to other BDLP structures, such as the structure of NpBDLP that is most closely related to fusion DLPs, the here presented SynDLP oligomer structure resembles a fission DLP structure, similar to classical eukaryotic Dynamin or Drp1, that has not been observed in bacteria before..." Could the authors elaborate what the resemblance and differences compared to the fission DLP, NpBDLP? A gallery comparing the structures might be helpful.
As suggested, we now present and compare structures of different DLPs in a new Fig. 2 and in Supplementary Fig. 8+10, involving the bacterial NpBDLP and MslniA as well as, in accordance with reviewer #2 point 2, Dynamin, MxA and Drp1. The aligned structures revealed closer relation of SynDLPs stalk and BSE domain with the eukaryotic DLPs confirming the eukaryotic structural features of the cyanobacterial SynDLP.

- 11. Intramolecular disulfide bridge in the BSE domain:
The intramolecular disulfide bridge and its conservation in the cyanobacterial KGK clade is intriguing and suggests a potential significance. This is an interesting observation, but it would benefit from a more thorough explanation of the author's observations. While there is a difference in GTPase activity in Figure 3g, the difference seems very low (and the activity still very high) compared to the >100-fold change in GTPase activity upon DLP oligomerization when binding to membranes. Given the reducing environment of the cytosol, it would be helpful to know more about the redox environment of SynDLP to understand if formation of this disulfide bond is reasonable *in vivo*.
In cyanobacteria, disulfide bond formation in the cytosol is mediated by Thioredoxins, as reported in the literature. We added this information and relevant references to the manuscript, and now write on page 23, lines 485-488: "In fact, several proteins in the cyanobacterial cytoplasm contain disulfide bridges, and the (in part reversible) formation of disulfide bridges is mediated by the Thioredoxin system⁷³⁻⁷⁵. Thus, it might even be possible that a reversible formation of the disulfide bridge in the BSE domain is involved in the regulation of the SynDLP activity."

- 12. In the discussion, "...It remains to be shown how SynDLP oligomerization and futile GTP hydrolysis might be prevented *in vivo*, likely by accessory proteins..." The authors assume here that the GTP hydrolysis *in vivo* is futile, however this reviewer is not convinced there is sufficient evidence to conclusively reach this conclusion. The authors also raise the idea of accessory proteins. It may be worthwhile here to comment on the region/residues in synDLP that senses negatively charged lipids, or what role a potential adaptor protein might play if there are no obvious residues for this type of interaction.
Based on the now available structure, we have identified two putative lipid binding regions (see comments to reviewers #1 and #2). We mutated the respective residues and analyzed any changes in the membrane-binding propensity of the isolated proteins, however, observed no clear changes in membrane interaction of the mutant proteins compared to the wt. We added the results to the manuscript in a new Supplementary Fig. 14 and now write on page 16-17, lines 324-329: "In the here solved SynDLPs structure, we recognized two putative MIDs at the tip of the stalk based either on the quaternary (Fig. 1h, Supplementary Fig. 14a) or the tertiary structure (Supplementary Fig. 14b). However, when these regions were mutated, the

isolated recombinant proteins still interacted with DOPG containing LUVs similar to SynDLP wt (Supplementary Fig. 14c+d), and thus these regions (alone) are not responsible for membrane binding of SynDLP. Potentially, either other regions of the protein or a larger area involving multiple SynDLP parts are responsible for membrane interaction.”

Reviewer #4:

Summary:

The authors report elegant structural, biochemical, and biophysical experiments that support a role for SynDLP, a dynamin-like protein from cyanobacterium Synechocystis, in fusing membranes. SynDLP forms oligomers in solution, in the absence of nucleotides or lipids. A 3.7 Å structure of the SynDLP oligomer was determined by cryo-electron microscopy. Similar to other dynamin like proteins, three interfaces enable oligomerization of the protein. Interestingly, a unique intramolecular disulfide bridge was observed in the bundle signaling element, BSE, which was determined to be important for the GTPase activity of SynDLP. SynDLP was found to bind negatively charged lipids of the thylakoid membrane, specifically, phosphatidylglycerol, and facilitates membrane fusion potentially by membrane destabilization via intercalation. This work presents a dynamin like protein, SynDLP, with unique features and mechanisms of membrane fusion by dynamin like proteins that would be fascinating to explore in future studies.

Minor concerns:

- 1. Could the authors comment on the essentiality of SynDLP in the lifecycle of Synechocystis? Or provide evidence of defective membrane repair in the cyanobacteria with mutant or knocked out SynDLP?

While we did not study the generated KO strain in detail yet, we can safely say that the gene/protein is not essential. We added this information to the legend to Supplementary Fig. 1 and now write in the legend, lines 32-35: “As the slr0869 gene could be completely deleted, the native SynDLP protein appears to be non-essential for the cyanobacterium, at least under the chosen growth conditions. The deletion strain grows like the wt and did not show severe defects in photosynthetic performance. Thus, our initial in vivo experiments indicated no altered phenotype of the SynDLP knock-out strain, at least under standard growth conditions.”

- 2. Could the authors identify putative lipid binding sites/residues in SynDLP? ***Inspection of the structure allowed us to identify two “paddle” regions putatively involved in membrane binding (now highlighted in Fig. 1h and Supplementary Fig. 14a+b). Membrane binding of mutants, where these putative membrane binding sites were mutated, were analyzed (as mentioned in the response to Reviewers #1, 2&3). We present the new data now in the revised manuscript on page 16-17, lines 324-329 as well as in Supplementary Fig. 14c+d, showing that the putative binding sites are not (alone) responsible for membrane interaction.***

- 3. Membrane fusion dynamins typically have a transmembrane region, with the fusion dynamin proteins bridging opposing membranes. Could the authors discuss or propose a mechanism by which SynDLP accomplishes membrane fusion without a clearly identified transmembrane domain? Or, Could the authors show electron micrographs of SynDLP organized on liposomes (LUV)? It will be informative to see how the protein is organized. That is are the liposomes tethered, tubulated, or destabilized? **We added DLS measurements of SynDLP incubated with liposomes to further support the idea of in vitro membrane fusion. Indeed, the measurements indicated the formation of larger structure and are now shown in Fig. 7a. However, in line with the SFG spectroscopy measurements (Fig. 5c+d), we likely observed membrane destabilization caused by intercalation of a putative SynDLP MID in the membrane, which results in membrane fusion under the chosen in vitro conditions. As the structure indicates SynDLP being a fission DLP and membrane destabilization is required for both, membrane fusion as well as fission, we toned down the statements concerning SynDLPs fusogenic activity. Yet, we updated the discussion about SynDLPs membrane activity and now write on page 25, lines 538-541: “Yet, SynDLP interaction with liposomes may result in membrane destabilization, a process needed for both membrane fusion as well as fission. A membrane destabilizing activity is well conceivable due to the observation that SynDLP not only binds to but intercalates into PG containing membranes (Fig. 5c), which induces perturbations in the lipid structure.”**

- 4. The authors suggest SynDLP is unique among dynamin like proteins in not requiring nucleotides or lipids to form oligomers. However, dynamin, the founding member, is also observed to form oligomers in the absence of lipids or nucleotides (PMID: 7877694). **We completely agree with this reviewer, yet in the specified paragraph we solely refer to bacterial dynamin-like proteins. Here, the oligomer properties of SynDLP are unique compared to other bacterial DLPs. Nevertheless, we added more information about dynamins oligomeric state on page 22, lines 456-463 and write: “Isolated SynDLP forms oligomers of approx. 40–50 molecules in solution in the absence of nucleotides and/or membranes (Fig. 1a+b). Nucleotide- and membrane-independent oligomerization has been reported for eukaryotic DLPs, such as Dynamin¹⁰. However, this is a unique feature not described in the field of bacterial DLPs thus far^{35,39,41–43}. Typically, DLPs oligomerize upon nucleotide-binding or upon binding to membrane surfaces¹⁸, and EM micrographs of the here observed SynDLP oligomers (Fig. 1b) looked, in fact, very similar to lipid-free oligomers formed by human Drp1 in the presence of GTP⁵⁰. However, in contrast to Drp1, SynDLP oligomers assemble already in the absence of nucleotides.”**

- 5. SynDLP is a GTPase, and it would be informative to show how its membrane fusion activity is affected as it proceeds through the GTPase cycle of nucleotide binding and hydrolysis. **We now performed the fusion assay in the presence of nucleotides and did not see any changes in the fusion activity. We added this information to the manuscript on page 21, lines 426-427 and present the data in Supplementary Fig. 20.**

- 6. How is the SynDLP oligomer's assembly and disassembly regulated? The current oligomeric structure was under low ionic conditions ~ 7 mM salt. How does nucleotide binding and hydrolysis affect assembly of the oligomer? Could the authors discuss what triggers SynDLP, assembly that presumably occurs during membrane repair. **To address this question, we now performed a sedimentation assay showing that oligomerization is only slightly altered in presence of different nucleotides. We now show these data in Supplementary Fig. 3 and write on page 6, lines 110-112: "In a sedimentation assay, the addition of GTP or GDP led to only marginal changes in the sedimentation behavior of SynDLP, indicating only a minor shift to larger structures in the presence of GTP and to smaller structures after GDP addition (Supplementary Fig. 3)." We also performed negative stain EM micrographs to investigate the influence of different NaCl concentrations showing longer and more ordered SynDLP filaments at 0 mM NaCl, which get shortened and less ordered with increasing NaCl concentrations. We added the micrographs in a new Supplementary Fig. 4.**

- 7. The quality of the structural data is excellent. However, could the authors comment on the presence of rotamer outliers? **The presence of rotamer outliers is indicated in Table 3 to be a proportion of 0.31%, which is according to MolProbity and the PDB validation report within the acceptable range compared to the percentile relative to all structures**

REVIEWERS' COMMENTS

Reviewer #1 (Remarks to the Author):

The authors have addressed all my points in their revision. This paper is a valuable addition to the literature about bacterial dynamins with a highly significant impact. I have no further questions/suggestions. Congratulations to all authors for this great work.

Reviewer #2 (Remarks to the Author):

The manuscript has been improved by the revisions. I am satisfied with the additional experiments but there are still some issues with the text and presentation:

Title: 'Uncommon activation' and Abstract: 'Such atypical GTPase domain interfaces might be a GTPase activity regulating tool in oligomerized SynDLP.'

As mentioned before, I still do not see any evidence of an uncommon activation mechanism in SynDLP - in fact, just the opposite - the new experiments appear to indicate a typical G domain dimerization-dependent mechanism. There is also little evidence for GTPase regulation by the new interface - the assembly-deficient mutant cannot be used to analyse the role of the GTPase-BSE interface, since oligomerization may affect GTPase activity in other ways, for example, by affecting dimerization of the GTPase domains in the dimer vs. the oligomer (also as mentioned before). Besides, GTPase activities of wt and interface-3 mutants are very similar. Without targeted mutations in the GTPase domain-BSE interface, these statements are too speculative and should be removed from title and abstract.

Same comment for line 567: The presence of such GD interfaces in SynDLP oligomers is critical for the relatively high basal GTPase activity and would, thus, be a novel concept of DLP trans-activation.

I do not believe that this is correct.

Abstract: The structural characteristics of SynDLP oligomers suggest it to be a bacterial ancestor of eukaryotic Dynamamin-like proteins.

Since BDLPs are all somehow related to dynamin, it should be clearly stated here that SynDLP is 'the closest known bacterial ancestor of dynamin'

Line 127: Interestingly, the curvature of the SynDLP oligomers is rather low compared to similar assemblies of other DLPs.

Compare the curvature directly to that of dynamin, since e.g. Drp1 has a similar curvature of its oligomer as SynDLP and many other DLPs do not have apparent curvature in their assembly at all. The curvature could also be indicated in Fig. 1e.

Fig. 1f: This is not a very useful topology plot, see e.g. here for a better example:

https://www.researchgate.net/figure/Structures-of-the-cytosolic-domain-of-Sey1p-A-Scheme-showing-the-domains-of-Sey1p-from_fig1_282036215

Such topology will help to better understand the architecture of this new protein, in particular the interplay of BSE1, BSE2 and BSE3 and the stalk helices (see also comment below to the GTPase domain comparison). Please add.

Fig. 1h. Some of the labels are still too small and cannot be properly recognized.

Line 149: Overall, based on the secondary structure assignments, the monomer consists of 62% alpha-helices and 5% sheets.

I do not believe this information is important in the main text.

Supplementary Fig. 6 would be more informative as a sequence alignment of related dynamin sequences.

Supplementary Fig. 8: Nice figure, but why not increase the size of the structural figures/oligomers?

Fig. 2, Supplementary Fig. 9. The same color code should be applied for these two related figures. If the color code of Fig. 2 is maintained, please add a color scale to the figure (similar as in Fig. S9). If a sequence alignment was added in Suppl. Figure 6, conservation could be directly shown there and Suppl. Fig. 9 could be removed. I would urgently suggest to increase the size of all structural representations in Fig. 2 to a maximum, since they are very hard to see. Please add labels for interfaces-1, 2 and 3. Which 150 sequences have been selected for the surface conservation plot? Ideally, only dynamin, MxA and Drp1 homologues should be used since this would indicate possible conservation of assembly interfaces in the stalk in SynDLP; these interfaces are obviously not conserved in other DLPs. In fact, if Supplementary Fig. 6 was an alignment, it could be well used to calculate conservation in the surface conservation plot.

Fig. 2, Supplementary Fig. 10. This is a useful comparison showing close relation of dynamin and SynDLP domains, but rmsd values should be extended to the GTPase domain as well, e.g. add a table with rmsd values for all three domains, including the number of aligned residues for each comparison.

Fig. 2b: The SynDLP GTPase domain seems almost 100 residues longer than the other GTPase domains, is this correct? If yes, what is the basis for this extension, e.g. are there additional (new) elements in the SynDLP GTPase domain? Again, a nice topology plot/comparison would help to better understand the new structure.

Line 229: As both proteins showed no residual (measurable) GTPase activity at 0 mM GTP (negative control without nucleotide; first data point in each curve) although a GTP regenerating system is added, we conclude that no significant amount of GTP/GDP has been co-purified with the proteins.

This is not a good argument. For example, the small GTPase Ras co-purifies with GDP, but would certainly not show GTPase activity in the presence of a regenerative system, since GDP is not released from the protein to be regenerated. Could the OD₂₆₀/280 ratio of the SynDLP preparation be a better argument? A value below 0.8 would indicate that no nucleotide is bound.

Fig. 3a. Please label the different helices directly in the figure - pink and magenta is not so easy to distinguish.

Reviewer #3 (Remarks to the Author):

Thank you for all your in preparing your rebuttal. Our concerns have been addressed.

Reviewer #4 (Remarks to the Author):

It is helpful that the revised manuscripts now states that the SynDLP protein appears to be non-essential in cyanobacteria under the current growth conditions.

The manuscript is strengthened by the discussion on putative lipid binding sites in SynDLP. These sites are shown in Fig 1h and Suppl. Fig 14 a-b. While mutations to this putative lipid bind didn't abrogate lipid interaction, it is informative to see these experiments. It also raises the possibility that

oligomerization of SynDLP could enhance the lipid binding, and mutations in the paddle domain and oligomerization interfaces could reduce the lipid interaction.

It is helpful that the authors now include DLS studies showing formation of larger liposomes in the presence of SynDLP. This experiment, in Fig 7a, provides additional evidence that SynDLP alone is able to facilitate membrane fusion. The SFG spectroscopy measurements do support a role for SynDLP in destabilizing membranes, a process that would precede membrane fission. It would be informative to see electron micrographs of wild type SynDLP or the various disulfide, oligomerization, and putative lipid binding mutants in the presence of liposomes, perhaps in a future study.

The addition, in page 22, discussing nucleotide- and membrane- independent oligomerization provides the reader with the appropriate context on the mechanisms of dynamin like proteins.

It is unique indeed that the nucleotide state does not affect the fusion activity of SynDLP in Suppl. Fig 20. The nucleotide state did have marginal effect in the oligomerization of SynDLP, Suppl. Fig 3.

The author's explanation about the rotamer outliers is acceptable as long as the outliers fit the density.

Response to reviewer's comments

We thank reviewers #1, 3 and 4 for their positive response and for approving our manuscript.

Reviewer #2

The manuscript has been improved by the revisions. I am satisfied with the additional experiments but there are still some issues with the text and presentation:

- *Title: 'Uncommon activation' and Abstract: 'Such atypical GTPase domain interfaces might be a GTPase activity regulating tool in oligomerized SynDLP.' As mentioned before, I still do not see any evidence of an uncommon activation mechanism in SynDLP - in fact, just the opposite - the new experiments appear to indicate a typical G domain dimerization-dependent mechanism. There is also little evidence for GTPase regulation by the new interface - the assembly-deficient mutant cannot be used to analyse the role of the GTPase-BSE interface, since oligomerization may affect GTPase activity in other ways, for example, by affecting dimerization of the GTPase domains in the dimer vs. the oligomer (also as mentioned before). Besides, GTPase activities of wt and interface-3 mutants are very similar. Without targeted mutations in the GTPase domain-BSE interface, these statements are too speculative and should be removed from title and abstract.*

We thank this reviewer for the assessment of SynDLP's activation mechanism and slightly toned down the statements concerning this topic. We removed the statement "uncommon activation" from the title and now entitle the manuscript "SynDLP is a dynamin-like protein of *Synechocystis* sp. PCC 6803 with eukaryotic features". Furthermore, we now write on page 2, line 25-26: "Beneath typical GD-GD contacts, such atypical GTPase domain interfaces might be a GTPase activity regulating tool in oligomerized SynDLP."

- *Same comment for line 567: The presence of such GD interfaces in SynDLP oligomers is critical for the relatively high basal GTPase activity and would, thus, be a novel concept of DLP trans-activation. I do not believe that this is correct.*

Again, we toned down the statements about the activation of SynDLP and now write on page 26, lines 537-538: "Taken together, while intramolecular GD-BSE1 interactions are described for other DLPs^{27,57}, the SynDLP structure and the analysis of an assembly-defective mutant indicate an additional role of the BSE domain for GTPase activation." and on page 28, lines 580-582: "The presence of such GD interfaces in SynDLP oligomers illustrates a distinctive concept for regulating the basal GTPase activity and would, thus, indicate a so far unique role of the BSE domain in a DLP."

- *Abstract: The structural characteristics of SynDLP oligomers suggest it to be a bacterial ancestor of eukaryotic Dynamin-like proteins. Since BDLPs are all somehow related to dynamin, it should be clearly stated here that SynDLP is 'the closest known bacterial ancestor of dynamin'*

We thank the reviewer for this comment. We completely agree and now write on page 2, lines 28-29: "The structural characteristics of SynDLP oligomers suggest it to be the closest known bacterial ancestor of eukaryotic dynamin.", on page 10, lines 178-179: "Thus, SynDLP is the closest known bacterial ancestor of a class of eukaryotic DLPs, such as dynamin or MxA." And on page 24, lines 482-484: "Detailed structural comparisons between SynDLP and bacterial and eukaryotic DLPs demonstrate a close relationship between SynDLP and eukaryotic representatives, indicating SynDLP being the closest known bacterial ancestor of eukaryotic dynamin, Drps and Mx proteins".

- *Line 127: Interestingly, the curvature of the SynDLP oligomers is rather low compared to similar assemblies of other DLPs. Compare the curvature directly to that of dynamin, since e.g. Drp1 has a similar curvature of its oligomer as SynDLP and many other DLPs do not have apparent curvature in their assembly at all. The curvature could also be indicated in Fig. 1e.*

We added more information on the curvature radius of other DLP assemblies and now write on page 6, lines 128-131: "The curvature radius of 50 nm of the SynDLP oligomers is rather high compared to similar assemblies of other DLPs that typically assemble in the presence of membranes and/or nucleotides. Here, the curvature radius is usually in the range of 13 to 26 nm (reviewed here²⁰). However, other DLP assemblies also show a low curvature comparable to SynDLP, e.g., Drp1⁵⁰."

- *Fig. 1f: This is not a very useful topology plot, see e.g. here for a better example: https://www.researchgate.net/figure/Structures-of-the-cytosolic-domain-of-Sey1p-A-Scheme-showing-the-domains-of-Sey1p-from_fig1_282036215. Such topology will help to better understand the architecture of this new protein, in particular the interplay of BSE1, BSE2 and BSE3 and the stalk helices (see also comment below to the GTPase domain comparison). Please add.*

As suggested, we added a topology plot in the representation as new Fig. 1f.

- *Fig. 1h. Some of the labels are still too small and cannot be properly recognized.*

As suggested, we increased the size of the labels in Fig. 1h.

- *Line 149: Overall, based on the secondary structure assignments, the monomer consists of 62% alpha-helices and 5% sheets. I do not believe this information is important in the main text.*

We agree with the reviewer that this information is not necessary and deleted the relevant text passage.

- *Supplementary Fig. 6 would be more informative as a sequence alignment of related dynamin sequences.*

We mention in the main text on page 3, lines 49-50, that “DLP family members are typically not highly conserved on the sequence level, with the exception of the GTPase domain (GD)”. An alignment of the *SynDLP* sequences with the structurally compared DLPs would therefore only indicate the conserved GD motifs and provides no further information. Thus, we refrain from adding a sequence alignment to Supplementary Fig. 6.

- *Supplementary Fig. 8: Nice figure, but why not increase the size of the structural figures/oligomers?*

As suggested, we now increased the structural representation of the oligomers.

- *Fig. 2, Supplementary Fig. 9. The same color code should be applied for these two related figures. If the color code of Fig. 2 is maintained, please add a color scale to the figure (similar as in Fig. S9). If a sequence alignment was added in Suppl. Figure 6, conservation could be directly shown there and Suppl. Fig. 9 could be removed. I would urgently suggest to increase the size of all structural representations in Fig. 2 to a maximum, since they are very hard to see. Please add labels for interfaces-1, 2 and 3. Which 150 sequences have been selected for the surface conservation plot? Ideally, only dynamin, MxA and Drp1 homologues should be used since this would indicate possible conservation of assembly interfaces in the stalk in *SynDLP*; these interfaces are obviously not conserved in other DLPs. In fact, if Supplementary Fig. 6 was an alignment, it could be well used to calculate conservation in the surface conservation plot.*

We edited Fig. 2 as suggested. Therefore, the color code of the surface conservation plot in Fig. 2 was updated to match with the color code used in Supplementary Fig. 9. We increased the size of all structural representations in Fig. 2 and added the positions of the oligomerization interfaces 1-3 in one side view of the surface conservation plot. The 150 sequences were selected by the online tool ConSurf and are *SynDLP*-related DLP sequences based on genomic data and typically not from structurally/biochemically characterized DLPs. A sequence alignment only with dynamin, MxA and Drp1 would not be beneficial, as the conservation on the sequence level is too low (especially for regions outside of the GD, e.g., the assembly interfaces) and the described similarities between the proteins were exclusively identified by structural/biochemical analyses. Thus, we refrain from the addition of a primary sequence alignment (as discussed above).

- *Fig. 2, Supplementary Fig. 10. This is a useful comparison showing close relation of dynamin and SynDLP domains, but rmsd values should be extended to the GTPase domain as well, e.g. add a table with rmsd values for all three domains, including the number of aligned residues for each comparison.*

As suggested, we added a table to the Supplementary Fig. 10 that now includes rmsd values of structural alignments of the GDs as well as the number of aligned residues used for each comparison.

- *Fig. 2b: The SynDLP GTPase domain seems almost 100 residues longer than the other GTPase domains, is this correct? If yes, what is the basis for this extension, e.g. are there additional (new) elements in the SynDLP GTPase domain? Again, a nice topology plot/comparison would help to better understand the new structure.*

We addressed this concern and now write on page 10, lines 170-171: “In fact, the SynDLP GD is >100 aa larger than typical dynamin-like GDs as it contains additional loops and α -helices”. We also added topology plots of the GD from SynDLP and two other DLP representatives to the new Supplementary Fig. 10a to better show the differences between the structures.

- *Line 229: As both proteins showed no residual (measurable) GTPase activity at 0 mM GTP (negative control without nucleotide; first data point in each curve) although a GTP regenerating system is added, we conclude that no significant amount of GTP/GDP has been co-purified with the proteins. This is not a good argument. For example, the small GTPase Ras co-purifies with GDP, but would certainly not show GTPase activity in the presence of a regenerative system, since GDP is not released from the protein to be regenerated. Could the OD₂₆₀/280 ratio of the SynDLP preparation be a better argument? A value below 0.8 would indicate that no nucleotide is bound.*

As suggested, we added a comment on the A_{260}/A_{280} ratio of purified SynDLP to further strengthen our statement that the purified protein is nucleotide-free. We now write on page 13, lines 237-238: “This is supported by the low A_{260}/A_{280} ratio of 0.7 determined via absorption spectroscopy using purified SynDLP, a value indicating that no nucleotides were bound.”

- *Fig. 3a. Please label the different helices directly in the figure - pink and magenta is not so easy to distinguish.*

As suggested, the different BSE helices 1-3 are now directly labeled in the figure.